# NBSP: A Neuron-Level Framework for Balancing Stability and Plasticity in Deep RL

## Abstract

In contrast to the human ability to continuously acquire knowledge, agents struggle with the stability-plasticity dilemma in deep reinforcement learning (DRL), which refers to the trade-off between retaining existing skills (stability) and learning new knowledge (plasticity). Current methods focus on balancing these two aspects at the network level, lacking sufficient differentiation and fine-grained control of individual neurons. To overcome this limitation, we propose Neuron-level Balance between Stability and Plasticity (NBSP) method, by taking inspiration from the observation that specific neurons are strongly relevant to task-relevant skills. Specifically, NBSP first (1) defines and identifies RL skill neurons that are crucial for knowledge retention through a goal-oriented method, and then (2) introduces a framework by employing adaptive gradient masking and experience replay techniques targeting these neurons to preserve the encoded existing skills while enabling adaptation to new tasks. Numerous experimental results on the Meta-World, Atari, and DMC benchmarks demonstrate that NBSP significantly outperforms existing approaches in balancing stability and plasticity.

## 1 Introduction

**Deep reinforcement learning (DRL)** has shown exceptional capabilities across a range of complex scenarios, such as gaming (Mnih et al., 2013), robotic manipulation (Andrychowicz et al., 2020), and autonomous driving (Kiran et al., 2021). However, most RL research focuses on agents that learn to solve individual problems rather than learn a sequence of tasks continually. Ideally, the agent must maintain its performance on previously learned tasks, referred to as **stability** (McCloskey & Cohen, 1989), while simultaneously adapting to new tasks, known as **plasticity** (Carpenter & Grossberg, 1987). However, it has been revealed that emphasizing stability may hinder the ability of agents to learn new knowledge (Nikishin et al., 2022a), whereas excessive plasticity can lead to catastrophic forgetting of previously acquired knowledge (Atkinson et al., 2021b), a challenge known as the **stability-plasticity dilemma** (eMermillod et al., 2013). Although there is a growing body of continual reinforcement learning (CRL) research (Wolczyk et al., 2022), our study specifically targets the intrinsic balance between stability and plasticity, which remains a fundamental and under-explored problem in DRL, and further advances this direction by exploring it at the neuron level.

Existing methods to strike a balance between stability and plasticity generally fall into three categories, i.e. (1) **regularization-based methods** (Kirkpatrick et al., 2017; Kumar et al., 2023), which apply penalties to parameter changes to mitigate forgetting while acquiring new knowledge; (2) **replay-based methods** (Ahn et al., 2024), which leverage past experiences to consolidate knowledge; and (3) **modularity-based methods** (Kim et al., 2023; Anand & Precup, 2024), which seek to decouple stability and plasticity or isolate different components for different tasks. Despite their contributions, these methods suffer from three key limitations: (1) They primarily operate at the network level, yet their ultimate impact manifests at the level of individual neurons. However, these methods fail to differentiate and fine-grained control neurons based on their specific roles. Therefore, identifying and effectively utilizing task-relevant neurons remains both critical and under-explored. (2) These studies are primarily conducted within the paradigm of continual learning, thus overlooking the unique characteristics intrinsic to DRL. (3) These approaches could sometimes unnecessarily inflate model parameters, thereby introducing unwarranted complexity (Bai et al., 2023).

By analyzing the activations of neurons in the DRL network, we observe that the activations of certain neurons are strongly correlated with the task goal. For instance, Figure 1 illustrates the activation distribution of a specific neuron in the network following training on the drawer-open task from the Meta-World benchmark (Yu et al., 2020). Activation of the neuron serves as a reliable predictor of whether the task is successful. Higher activation levels correspond to an increased likelihood of completing the task successfully, indicating that this neuron encodes a critical skill essential for the task. Consequently, it plays a pivotal role in retaining task-specific memory.

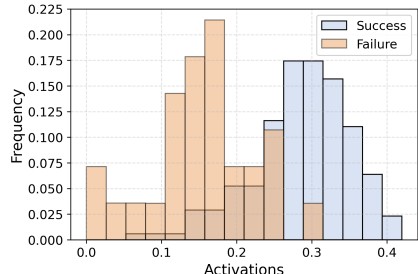

Figure 1: Distribution histogram of the activation of a neuron, categorized based on whether the drawer-open task was successfully completed or not.

Motivated by the aforementioned observations, we propose **Neuron-level Balance between Stability and Plasticity (NBSP)**, a novel DRL framework that operates at the level of neurons to tackle the stability-plasticity dilemma. In particular, (1) we first introduce **RL skill neurons**, which encode critical skills necessary for knowledge retention. While skill neurons have been investigated and successfully exploited in various domains, such as pre-trained language models (Wang et al., 2022) and neural machine translation (Bau et al., 2018), skill neurons are still much less explored in DRL. We bridge this research gap by proposing a goal-oriented strategy for identifying RL skill neurons. (2) We then apply adaptive **gradient masking** according to the scores of these neurons, ensuring that the encoded knowledge of prior skills is preserved while allowing fine-tuning during subsequent training. Meanwhile, the other neurons retain the ability to learn new tasks. (3) Additionally, we incorporate **experience replay** to periodically revisit the past experience to reinforce stability, preventing excessive drift from previous knowledge. Integrally, NBSP offers three key advantages compared with previous methods: (1) The neuron-level processing enables finer control and greater flexibility, addressing the stability-plasticity trade-off at the most fundamental level of the network. (2) The goal-oriented approach for identifying RL skill neurons is specifically tailored to DRL. (3) This framework is simple, avoiding complex network designs or additional parameters.

We conduct experiments on the **Meta-World** (Yu et al., 2020), **Atari** (Mnih et al., 2013), and **DMC** (Tunyasuvunakool et al., 2020) benchmarks to evaluate the effectiveness of NBSP. Our results show that NBSP outperforms existing methods in balancing stability and plasticity, enabling effective learning of new tasks while preserving knowledge from previous tasks. Additionally, we perform extensive ablation studies to investigate the contribution of different components within NBSP. Specially, we analyze the DRL agents by dissecting the performance of the two critical modules, i.e., the actor and the critic. Our findings reveal that (1) addressing both the actor and critic networks yields the best performance, and (2) the critic plays a more critical role in achieving this balance due to the differences in their inherent training mechanisms. In summary, our key contributions include:

- We introduce the concept of RL skill neurons which encode skills of the task, essential for knowledge retention, and propose a goal-oriented strategy specifically tailored to DRL for identification.
- We tackle the stability-plasticity dilemma in DRL from the perspective of RL skill neurons, by employing gradient masking and experience replay on these neurons, eliminating requirements of complex network designs or additional parameters.
- We conduct extensive experiments on the Meta-World, Atari, and DMC benchmarks to demonstrate the effectiveness of our method in balancing stability and plasticity.

## 2 RELATED WORK

**Balance between stability and plasticity**. In DRL, addressing the stability-plasticity dilemma (Carpenter & Grossberg, 1988) has inspired various strategies. Stability-focused methods often utilize replay techniques, such as A-GEM (Chaudhry et al., 2018b), using episodic memory to constrain loss, and ClonEx-SAC (Wolczyk et al., 2022), enhancing performance through behavior cloning. Plasticity-focused methods aim to preserve network expressiveness, with solutions like CBP (Dohare et al., 2024), resetting (Nikishin et al., 2022b), plasticity injection (Nikishin et al., 2024), Reset & Distillation (Ahn et al., 2024), and CRelu (Abbas et al., 2023) to prevent activation collapse.

Modularity-based methods balance stability and plasticity by decoupling task-specific knowledge, exemplified by soft modularity for routing networks (Yang et al., 2020), value function decomposition (Anand & Precup, 2024), and compositional frameworks leveraging neural components (Mendez et al., 2022). Shrink & Perturb (Ash & Adams, 2020), which updates weights with shrinkage and noise, and UPGD (Elsayed & Mahmood, 2024), which varies update sizes by unit utility, also target the balance. Methods such as CRelu and ClonEx-SAC focus on continual reinforcement learning(CRL), but our study specifically targets the intrinsic balance between stability and plasticity, with other factors such as task order controlled in a cycling task setting. Moreover, while most methods operate at the network level, our approach explores neuron-level research, providing fine-grained control.

**Neuron-level research**. Recent research has shown that neuron sparsity often correlates with task-specific performance (Xu et al., 2024), driving a growing focus on skill neurons to interpret network behavior and tackle challenges across domains. For example, skill neurons have been used to enhance transferability and efficiency in Transformers via pruning (Wang et al., 2022). Other studies, such as identifying Rosetta Neurons (Dravid et al., 2023) and language-specific neurons (Tang et al., 2024), have advanced alignment and interpretability. Despite these achievements, the exploration of skill neurons in DRL remains largely under-explored. Some works focus on task-specific sub-network selection at the neuron level, such as CoTASP and PackNet. At a finer granularity, other approaches target individual neuron management. NPC identifies and constrains important neurons for maintaining stability. Similarly, ReDO (Sokar et al., 2023) and its successor GraMa (Liu et al., 2025) introduce schemes for dormant neuron management and activity quantification based on activation and gradient magnitudes. NE (Liu et al., 2024) dynamically adapts network topology through neuron growth and pruning based on potential gradients. However, these methods overlook the fundamental link between a neuron and the task's goal in RL, and identify neurons through static measures like activation or gradient. Our work addresses this gap and moves beyond the static measures by leveraging a significant correlation between a neuron activation and the specific objective of the RL task. This insight allows us to identify the underlying skill neurons, those that are functionally relevant to the task success.

## 3 METHODOLOGY

In this section, we first introduce the terminology of RL skill neurons and then propose the Neuron-level Balance between Stability and Plasticity (NBSP) method.

### 3.1 PROBLEM SETUP

We focus on the setting of sequential task learning without constraints on the time intervals between tasks. In this setting, the agent is expected to perform all previously learned tasks after training, without relying on task-specific signals. For instance, large models such as DeepSeek employ RL to enhance their reasoning capabilities. However, different tasks, such as vision and mathematics, demand distinct reasoning abilities. To first strengthen a specific type of reasoning and then generalize to others, it is essential to strike a balance between stability and plasticity during sequential training. Furthermore, in real-world applications, the enhanced model should be able to handle all tasks without relying on explicit task signals. Let $\tau \in \{\tau_1, \tau_2, ...\}$ represent a sequence of task, each task $\tau$ corresponds to a distinct Markov Decision Process (MDP) $M^\tau = (S^\tau, A^\tau, P^\tau, R^\tau, \gamma^\tau)$, where $S^\tau$, $A^\tau$, $P^\tau$, $R^\tau$ and $\gamma^\tau$ denote the state space, action space, transition dynamics, reward function, and discount factor, respectively. Instead of addressing a single MDP, the goal is to solve a sequence of MDPs one by one using a universal policy $\pi(a|s)$ and Q-function $Q(s, a)$. The primary challenge lies in balancing plasticity, which refers to maximizing the discounted return of the current task, and stability, which emphasizes the maximization of the expected discounted return averaged across all previous tasks. This trade-off constitutes the core problem addressed in this work.

### 3.2 IDENTIFYING RL SKILL NEURONS

In this study, we make a key observation that the stability and plasticity of the agent network are closely related to its expressive capabilities, which are significantly influenced by the behavior of individual neurons. As evidenced in Molchanov et al. (2022), neuron expression determines how information is propagated and processed, directly affecting the learning and knowledge retention capabilities of the network. Therefore, understanding and controlling neuron behavior is at the most

fundamental level for striking a balance between stability and plasticity. On the one hand, when neuron expression is stable and generalized, the agent network tends to exhibit high stability. On the other hand, strong plasticity can be achieved given neuron expression is flexible and adaptable.

Several works have demonstrated the multifaceted capabilities of neurons, such as the storage of factual knowledge (Dai et al., 2022), the association with specific languages (Tang et al., 2024), and the encoding of safety information (Chen et al., 2024). These specialized neurons, often referred as skill neurons, have been shown to significantly contribute to network performance (Wang et al., 2022). However, the potential of skill neurons in DRL remains largely under-explored. As illustrated in Figure 1, activations of the specific neuron are strongly correlated with task success: higher activation levels increase the likelihood of successful task completion, whereas lower levels are associated with failure. ***This indicates that the activations of these neurons significantly affect agent performance, effectively encoding the critical skills required for the task. By preserving the activations of such neurons, it becomes possible to retain the learned task-specific skills, thereby improving stability.***

In this work, we formally define these special neurons as **RL skill neurons**, which encode critical skills, essential for knowledge retention in DRL. Furthermore, we propose a goal-oriented method for the identification of these neurons. Unlike prior approaches that primarily focus on the inputs triggering neuron activations (Bau et al., 2020; Gurnee & Tegmark, 2023), our method emphasizes their impact on achieving ultimate goals, i.e. succeeding in finishing Meta-World tasks and attaining high scores in Atari games, by comparing the activation patterns of the neurons that exhibit varying performance levels. In Section 4.2, we empirically show the advantage of our goal-oriented method.

For a specific neuron $\mathcal{N}$, let $a(\mathcal{N}, t)$ represent its activation at step $t$. In fully connected layers, each output dimension corresponds to the activation of a specific neuron, whereas in convolution layers, the average of each output channel represents the activation of a neuron. To quantify activation level of a neuron $\mathcal{N}$, we define the **average activation** as:

$$\overline{a}(\mathcal{N}) = \frac{1}{T_{avg}} \sum_{t=1}^{T_{avg}} a(\mathcal{N}, t), \tag{1}$$

where $T_{avg}$ represents the step over which we compute the average activation. The neuron activation level can then be assessed by comparing its current activation with the average activation.

To assess the performance of the agent at step $t$, we introduce the **Goal Metric (GM)**, denoted as $q(t)$. It serves as an evaluation metric for assessing the performance of the agent's network, varying based on the objective of the task. It is computed in an online manner during training. For instance, on the Meta-World benchmark, the GM is typically binary, determined by whether the episode is successful, which is computed at the end of each episode. In contrast, the GM is determined by the cumulative return of the episode for the Atari benchmark. Additionally, we define the **average Goal Metric** (GM) of the agent as follows, which serves as a baseline for evaluating the performance by comparing it with the current GM.

$$\overline{q} = \frac{1}{T_{avg}} \sum_{t=1}^{T_{avg}} q(t). \tag{2}$$

To differentiate the roles of neurons across various tasks, it is essential to assess neuron activations in relation to specific goals. Intuitively, we can consider a neuron $\mathcal{N}$ to be positively contributing to the goal at step $t$ when its activation $a(\mathcal{N}, t)$ surpasses the average activation $\overline{a}(\mathcal{N})$, i.e. $a(\mathcal{N}, t) > \overline{a}(\mathcal{N})$, while the GM at the same step also exceeds its average, i.e. $q(t) > \overline{q}$. To quantify this contribution, we accumulate a batch of results over $T$ steps and define the over-activation rate as follows:

$$R_{over}(\mathcal{N}) = \frac{\sum_{t=1}^{T} 1_{[1_{[a(\mathcal{N}, t) > \overline{a}(\mathcal{N})]} = 1_{[q(t) > \overline{q}]}]}}{T}. \tag{3}$$

Here, $1_{[condition]} \in \{0, 1\}$ denotes the indicator function, which returns 1 if and only if the specified condition is satisfied. While Eq. (3) assesses the positive correlation of neurons towards achieving the goal, covering the cases $(a > \overline{a}, q > \overline{q})$ and $(a < \overline{a}, q < \overline{q})$, where activation and performance change in the same direction (positive correlation). However, it overlooks neurons that exhibit a negative correlation with the goal but still carry valuable task-related knowledge, covering the cases $(a < \overline{a}, q > \overline{q})$ and $(a > \overline{a}, q < \overline{q})$. Specifically, when the activation of a neuron falls below its

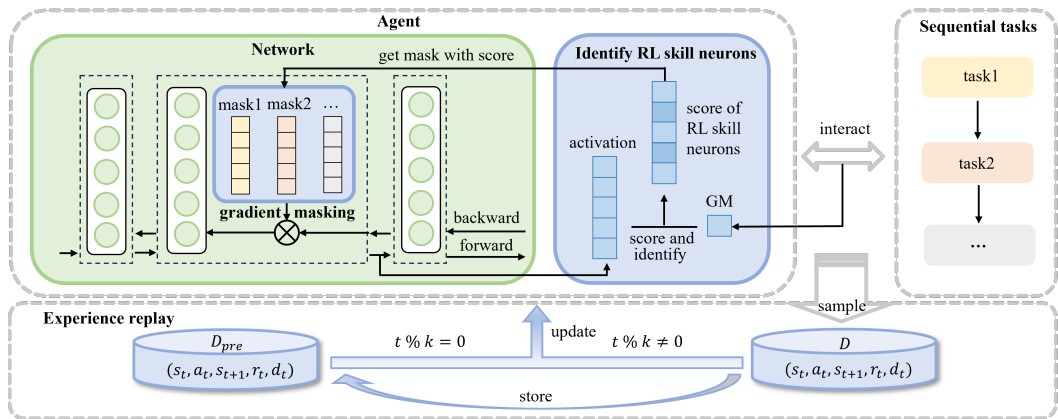

Figure 2: Framework of NBSP. The agent scores and identifies RL skill neurons for each task by measuring the activation in relation to the GM. While learning new tasks, the gradient of these neurons is masked adaptively based on their scores to preserve the encoded skills, while still allowing fine-tuning for new task learning. Additionally, a replay buffer is used to store a portion of the experiences from previous tasks, which is periodically sampled to update the agent.

average activation, the agent performs well conversely. To this end, we define a **comprehensive score** **Score**$(\mathcal{N})$ for the neuron that takes into account both positive and negative effects:

$$Score(\mathcal{N}) = max(R_{over}(\mathcal{N}), 1 - R_{over}(\mathcal{N})). \tag{4}$$

Subsequently, we rank all neurons in the agent network, excluding those in the last layer for the few neurons and their large role in determining performance, which is validated in Appendix C.10.5, in descending order based on their scores. The RL skill neurons are determined by selecting the neurons with the top m% highest scores, formally defined as follows, where $\tau_m(\cdot)$ denotes the top-m selection operator. And the pseudo-code of the identification method is shown in Appendix D.

$$\mathcal{N}_{RL\ skill} = \tau_m(Score(\mathcal{N})) \tag{5}$$

### 3.3 NEURON-LEVEL BALANCE BETWEEN STABILITY AND PLASTICITY

Building upon the concept of RL skill neurons, we propose a novel DRL framework — **Neuron-level Balance between Stability and Plasticity (NBSP)**, as shown in Figure 2. Unlike prior methods (Bai et al., 2023; Kim et al., 2023), the framework proposed does not require complex network designs or additional parameters. Given that RL skill neurons encode essential task-specific skills, preserving their activation patterns is critical to maintaining knowledge from previous tasks during continual tasks learning. However, simply freezing RL skill neurons would hinder the ability of the agent to adapt to new tasks. To address this challenge, NBSP employs an adaptive **gradient masking** technique. Specifically, during each update round in the continual learning process, the gradients of RL skill neurons are selectively masked to restrict changes in their activation patterns while allowing other neurons to adapt freely. This process is formally expressed as follows:

$$\Delta W_{:,j} = mask_j^{(l)} \cdot \Delta W_{:,j}^{(l)}, \tag{6}$$

where $\Delta W_{:,j}^{(l)}$ denotes the gradient with respect to the weight $W_{:,j}^{(l)}$ in the $l$-th layer of the network, and $j$ is the index of the output neuron in that layer. The term $mask_j^{(l)}$ is associated with the score of $j$-th neuron in the $l$-th layer, which could be calculated as follows:

$$mask(\mathcal{N}) = \begin{cases} \alpha(1 - Score(\mathcal{N})) & \text{if } \mathcal{N} \in \mathcal{N}_{RL\ skill} \\ 1 & \text{if } \mathcal{N} \notin \mathcal{N}_{RL\ skill} \end{cases}, \tag{7}$$

where $\mathcal{N}_{RL\ skill}$ represents the set of RL skill neurons, and $\alpha$ is a super-parameter that determines the degree of restriction on these neurons, which is configured to 0.2 in the experiment. And gradient masking is applied only to the online networks, where gradients are computed and parameters are updated. The target networks are updated using EMA from the masked online networks. No

additional masking operation is required for target networks. ***By employing gradient masking, NBSP effectively safeguards the encoded skills within RL skill neurons from interference during the learning of new tasks, thereby enhancing stability. At the same time, RL skill neurons remain adaptable, allowing fine-tuning to accommodate new tasks and maintaining high plasticity. In addition, neurons except RL skill neurons are free to fully engage in learning new task-specific knowledge, ensuring comprehensive learning across tasks.***

To mitigate excessive drift from knowledge acquired in previous tasks, we integrate the **experience replay** technique, periodically sampling prior experiences at specific intervals $k$. After training on a task, a portion of the experiences, rather than the entirety, are stored in a unified replay buffer $D_{pre}$, requiring only a modest memory footprint. By incorporating experience replay, the stability of DRL agents is further enhanced. The corresponding loss function is defined as follows:

$$\mathcal{L} = R(t) \cdot \mathbb{E}_{(s_t,a_t,s_{t+1},r_t,d_t) \sim D_{pre}}[L] + (1 - R(t)) \cdot \mathbb{E}_{(s_t,a_t,s_{t+1},r_t,d_t) \sim D}[L], \quad (8)$$

where $L$ denotes the original loss function, $R(t)$ is a binary function that evaluates to 1 if and only if the current step $t$ is at an interval. $D$ represents the replay buffer for the current task, and $(s_t, a_t, s_{t+1}, r_t, d_t)$ denotes the tuple of the current state, action, next state, reward, and whether the episode is done sampled from the replay buffer. The pseudo-code of NBSP is shown in Appendix D.

## 4 EXPERIMENT

In this section, we evaluate the performance of NBSP on the **Meta-World** (Yu et al., 2020), **Atari** (Mnih et al., 2013) and **DMC** (Tunyasuvunakool et al., 2020) benchmarks.

**Experiment setting**. We follow the the experimental paradigm of Abbas et al. (2023); Liu et al. (2024), evaluating NBSP on a **cycling sequence of tasks** characterized by non-stationarity due to changing environments over time. Specifically, the agent learns each task sequentially and transitions to the next without resetting the learned networks. The task cycles through a fixed sequence, with a cycle completing once all tasks in the sequence have been learned. The agent cycles twice, resulting in each task being repeated twice. Details about the benchmarks are shown in Appendix C.2. For all experiments, we use the Soft Actor-Critic (SAC) (Haarnoja et al., 2018) algorithm, as implemented by CleanRL (Huang et al., 2022). Each agent is trained until either reaching a predefined maximum number of steps or demonstrating stable mastery of the task in the Meta-World benchmark. Each experiment is repeated using three different random seeds. The shaded regions in the figures and the plus/minus numbers represent the standard error across multiple seeds. Detailed descriptions of the hyperparameters and other experimental settings are provided in Appendix C.3.

Compared to the CRL training paradigm, our cycling training paradigm provides a more specific evaluation of the balance between stability and plasticity. By repeating each task twice within a cycling sequence, the setup not only assesses the plasticity in adapting to new tasks but also evaluates its stability when revisiting previous tasks, mitigating the influence of task order.

**Metric**. Overall performance is commonly assessed using the **Average Success Rate (ASR)**, analogous to the AIA metric (Wang et al., 2024). The higher the ASR, the better the method balances stability and plasticity. To evaluate the stability of the agent, we utilize the **Forgetting Measure (FM)** (Chaudhry et al., 2018a). The lower the FM, the better the method maintains stability. To assess the plasticity of the agent, we employ the **Forward Transfer (FWT)** metric (Lopez-Paz & Ranzato, 2017). The higher the FWT, the better the method maintains plasticity. Further details about evaluation metrics are available in Appendix C.4.

**Baseline**. To assess the effectiveness of the NBSP framework, we compare it with nine baseline methods dealing with the balance between stability and plasticity. **EWC** (Kirkpatrick et al., 2017) and **NPC** (Paik et al., 2019) primarily emphasize maintaining stability, while **CRelu** (Abbas et al., 2023), **CBP** (Dohare et al., 2024), and **PI** (Nikishin et al., 2024) focus on enhancing plasticity. **ANCL** (Kim et al., 2023), **CoTASP** (Yang et al., 2023), **NE** (Liu et al., 2024) and **UPGD** (Elsayed & Mahmood, 2024) aim to achieve a balance between stability and plasticity. Notably, CoTASP makes relevant tasks share more neurons in the meta-policy network, NPC consolidates important neurons, and NE dynamically adapts network topology via neuron growth and pruning, they are all relevant to neurons. Detailed descriptions of these baselines can be found in Appendix C.1.

Table 1: Results of NBSP with other baselines on the Meta-World benchmark.

| Cycling sequential tasks | Metrics | Methods | | | | | | | | | |
|---|---|---|---|---|---|---|---|---|---|---|---|
| | | EWC | NPC | ANCL | CoTASP | CRelu | CBP | PI | NE | UPGD | NBSP |
| (window-open → window-close) | ASR ↑ | 0.63 ± 0.03 | 0.26 ± 0.01 | 0.66 ± 0.04 | 0.05 ± 0.01 | 0.26 ± 0.14 | 0.67 ± 0.05 | 0.61 ± 0.02 | 0.83 ± 0.04 | 0.65 ± 0.05 | **0.90 ± 0.04** |
| | FM ↓ | 0.89 ± 0.07 | 0.68 ± 0.04 | 0.84 ± 0.10 | **0.01 ± 0.01** | 0.66 ± 0.42 | 0.78 ± 0.13 | 0.91 ± 0.07 | 0.27 ± 0.12 | 0.81 ± 0.09 | **0.18 ± 0.01** |
| | FWT ↑ | **0.97 ± 0.02** | 0.26 ± 0.01 | 0.97 ± 0.03 | 0.04 ± 0.01 | 0.33 ± 0.19 | 0.95 ± 0.02 | 0.94 ± 0.01 | 0.95 ± 0.01 | 0.95 ± 0.02 | **0.96 ± 0.02** |
| (drawer-open → drawer-close) | ASR ↑ | 0.68 ± 0.06 | 0.35 ± 0.05 | 0.64 ± 0.02 | 0.07 ± 0.01 | 0.29 ± 0.20 | 0.61 ± 0.03 | 0.60 ± 0.07 | 0.72 ± 0.02 | 0.72 ± 0.01 | **0.96 ± 0.02** |
| | FM ↓ | 0.80 ± 0.15 | 0.69 ± 0.05 | 0.88 ± 0.09 | **0.01 ± 0.01** | 0.31 ± 0.32 | 0.91 ± 0.03 | 0.71 ± 0.30 | 0.60 ± 0.02 | 0.69 ± 0.02 | **0.07 ± 0.06** |
| | FWT ↑ | **0.98 ± 0.01** | 0.39 ± 0.09 | 0.96 ± 0.01 | 0.09 ± 0.00 | 0.42 ± 0.28 | 0.93 ± 0.04 | 0.88 ± 0.15 | 0.93 ± 0.01 | 0.96 ± 0.01 | **0.98 ± 0.01** |
| (button-press-topdown → window-open) | ASR ↑ | 0.66 ± 0.06 | 0.25 ± 0.00 | 0.61 ± 0.01 | 0.03 ± 0.00 | 0.33 ± 0.10 | 0.62 ± 0.01 | 0.63 ± 0.02 | 0.71 ± 0.03 | 0.51 ± 0.06 | **0.95 ± 0.05** |
| | FM ↓ | 0.85 ± 0.14 | 0.67 ± 0.00 | 0.95 ± 0.05 | **0.01 ± 0.00** | 0.94 ± 0.01 | 0.97 ± 0.03 | 0.97 ± 0.05 | 0.73 ± 0.10 | 0.68 ± 0.14 | **0.08 ± 0.12** |
| | FWT ↑ | 0.96 ± 0.01 | 0.25 ± 0.01 | 0.95 ± 0.03 | 0.04 ± 0.01 | 0.42 ± 0.20 | **0.98 ± 0.02** | **0.98 ± 0.02** | 0.96 ± 0.01 | 0.71 ± 0.13 | **0.98 ± 0.01** |
| (window-open → window-close → drawer-open → drawer-close) | ASR ↑ | 0.44 ± 0.05 | 0.19 ± 0.04 | 0.48 ± 0.04 | 0.04 ± 0.01 | 0.10 ± 0.06 | 0.43 ± 0.03 | 0.41 ± 0.06 | 0.61 ± 0.04 | 0.46 ± 0.01 | **0.66 ± 0.14** |
| | FM ↓ | 0.74 ± 0.11 | 0.50 ± 0.02 | 0.80 ± 0.04 | **0.04 ± 0.01** | 0.39 ± 0.02 | 0.91 ± 0.05 | 0.84 ± 0.05 | 0.55 ± 0.06 | 0.50 ± 0.03 | **0.48 ± 0.18** |
| | FWT ↑ | 0.83 ± 0.10 | 0.20 ± 0.05 | 0.89 ± 0.06 | 0.08 ± 0.01 | 0.13 ± 0.10 | **0.97 ± 0.02** | 0.82 ± 0.10 | 0.84 ± 0.08 | 0.71 ± 0.01 | **0.89 ± 0.12** |
| (button-press-topdown → window-close → door-open → drawer-close) | ASR ↑ | 0.43 ± 0.03 | 0.17 ± 0.01 | 0.44 ± 0.03 | 0.04 ± 0.01 | 0.14 ± 0.11 | 0.41 ± 0.02 | 0.38 ± 0.01 | 0.59 ± 0.04 | 0.34 ± 0.01 | **0.74 ± 0.07** |
| | FM ↓ | 0.81 ± 0.09 | 0.47 ± 0.01 | 0.87 ± 0.02 | **0.04 ± 0.00** | 0.62 ± 0.16 | 0.94 ± 0.02 | 0.97 ± 0.02 | 0.55 ± 0.01 | 0.59 ± 0.02 | **0.34 ± 0.15** |
| | FWT ↑ | 0.88 ± 0.10 | 0.19 ± 0.02 | 0.91 ± 0.08 | 0.07 ± 0.02 | 0.17 ± 0.15 | **0.97 ± 0.01** | 0.92 ± 0.07 | 0.91 ± 0.03 | 0.55 ± 0.02 | **0.95 ± 0.06** |

## 4.1 Experiment on the Meta-World Benchmark

The experimental results of NBSP compared with other baselines on the Meta-World benchmark are presented in Table 1. As shown in the final column, NBSP significantly outperforms all other methods in the overall performance metric ASR. For two-task cycling tasks, NBSP achieves an ASR consistently above 0.9, which is substantially higher than other baselines. Its stability metric, FM, is markedly lower, while its plasticity metric, FWT, remains at a high level. Furthermore, NBSP also demonstrates excellent performance in four-task cycling tasks, maintaining a substantial lead.

For stability-focused baselines, EWC achieves a relatively good ASR but still falls short of NBSP. Moreover, EWC exhibits poor stability due to its high FM values. NPC performs even worse, failing to maintain both stability and plasticity effectively. Among plasticity-focused baselines, CBP and PI achieve comparable plasticity to NBSP, as reflected in their high FWT scores. However, both suffer from severe stability loss, indicated by their higher FM values. Another plasticity-focused method, CRelu, underperforms in both stability and plasticity. For baselines attempting to balance stability and plasticity, ANCL achieves high plasticity with competitive FWT scores but fails to retain prior knowledge, as reflected by its high FM value. CoTASP, despite being explicitly designed for this trade-off, performs poorly overall. NE achieves the best metrics among baselines but still falls short of NBSP, while UPGD trails NE slightly yet remains competitive.

The effectiveness of NBSP is further demonstrated in Figure 3, which showcases the training dynamics of NBSP. Specifically, during the second cycle of learning the same task, the agent exhibits a high success rate even before retraining, indicating that it has retained significant task knowledge. As a result, the agent is able to master the task more rapidly. This highlights the ability of NBSP to preserve knowledge from prior tasks while simultaneously maintaining the plasticity required to learn new tasks effectively. The other training process is demonstrated in Appendix C.7. In summary, ***NBSP delivers a remarkable improvement in maintaining stability without compromising plasticity, achieving a well-balanced trade-off in DRL.***

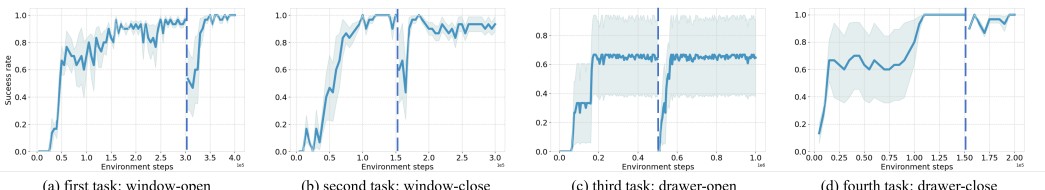

(a) first task: window-open  (b) second task: window-close  (c) third task: drawer-open  (d) fourth task: drawer-close

Figure 3: Training process of NBSP on the Meta-World benchmark. The segments to the left and right of the dashed line represent the training processes of the first and second cycles, respectively.

## 4.2 Ablation Study

To reveal the underlying working mechanisms of NBSP, we further evaluate (1) the two primary components of NBSP: the gradient masking technique and experience replay technique, (2) the neuron identification method, and (3) the two critical modules of DRL: the actor and the critic, (4)

sensitivity analysis of hyperparameters, (5) the role of experience replay, (6) the impact of task order, (7) the generalization of NBSP applied to the PPO algorithm (Schulman et al., 2017).

**Gradient masking and experience replay**. To evaluate the contributions of the two core components of NBSP, we designed five experimental settings: (1) vanilla SAC, (2) SAC with only the experience replay, (3) SAC with only the gradient masking, (4) SAC with experience replay and hard gradient masking, where the masks of RL skill neurons are set directly to zero, and (5) NBSP.

The results of the cycling sequential tasks (button-press-topdown → window-open) are shown in Table 2. From the results, we observe that: (1) The vanilla SAC algorithm suffers from severe stability loss, as indicated by a high FM score. (2) Using either experience replay or gradient masking alone alleviates the stability loss

Table 2: Results of ablation study of gradient masking and experience replay techniques.

| Metrics | (button-press-topdown → window-open) | | | | |
|---|---|---|---|---|---|
| | vanilla SAC | experience replay | only adaptive gradient masking | only hard gradient masking | NBSP |
| ASR ↑ | 0.62 ± 0.01 | 0.70 ± 0.08 | 0.71 ± 0.06 | 0.71±0.03 | **0.95 ± 0.05** |
| FM ↓ | 0.99 ± 0.02 | 0.50 ± 0.16 | 0.73 ± 0.21 | 0.72±0.04 | **0.08 ± 0.12** |
| FWT ↑ | **0.98 ± 0.02** | 0.92 ± 0.05 | 0.97 ± 0.02 | **0.98±0.03** | **0.98 ± 0.01** |

to some extent, confirming their individual effectiveness. (3) Combining both techniques in NBSP significantly improves performance, with lower FM and higher FWT. (4) Our adaptive gradient masking, which sets masks of RL skill neurons based on their scores, outperforms hard masking (setting masks to zero directly), demonstrating its superior effectiveness. ***These findings demonstrate that neither experience replay nor gradient masking alone can properly balance stability and plasticity, while their combination achieves optimal performance.*** The reason is that gradient masking and experience replay focus on different mechanisms and therefore complement each other. Gradient masking primarily targets RL skill neurons to reduce interference with past knowledge while maintaining the ability to fine-tune for new tasks. And experience replay mainly acts on neurons except RL skill neurons to prevent these neurons from being overly biased toward new tasks. Additional results for different task settings are provided in Appendix C.10.1.

To further assess the effectiveness of gradient masking on RL Skill neurons, we measure the sensitivity of network to perturbations applied to neurons that NBSP would mask (RL Skill neurons) versus those it would not (non–RL Skill neurons). In the button-press-topdown task, we inject zero-mean Gaussian noise into the activations of 10% of neurons. The noise is added either to (1) RL Skill neurons or (2) non–RL Skill neurons, yielding success rates of 0.22 ± 0.03 and 0.92 ± 0.02, respectively. Perturbing RL Skill neurons leads to a substantial performance drop, indicating that these neurons are critical for encoding task-specific knowledge. In contrast, perturbing non–RL Skill neurons has only a minor effect. This contrast demonstrates that masking RL Skill neurons is essential for preserving learned behavior, thereby supporting the stability side of the stability–plasticity trade-off.

To more clearly characterize the effect of experience replay, we conduct additional analyses on gradient-norm variance and parameter drift. (1) Gradient-norm variance. We report gradient-norm variance across four settings:

Table 3: Results of network performance with different component.

| | vanilla SAC | NBSP | SAC + gradient masking | SAC + experience replay |
|---|---|---|---|---|
| gradient-norm variance | 7.45 | 5.62 | 3.38 | 8.96 |
| network similarity | 0.56 | 0.89 | 0.79 | 0.75 |

vanilla SAC, SAC with gradient masking, SAC with experience replay, and NBSP, as shown in Table 3. The ordering of gradient-norm variance is SAC + experience replay > SAC > NBSP > SAC + gradient masking. This ordering reveals a fundamental limitation of experience replay: although replay introduces past transitions that help maintain memory, it also injects high-variance gradients due to stale, off-policy, and often task-mismatched samples. Such noisy gradients destabilize optimization and increase the likelihood of updates drifting toward suboptimal directions, particularly when the agent encounters new tasks. Consequently, experience replay alone can harm plasticity because its inherent gradient instability interferes with the efficient acquisition of new behaviors. In contrast, gradient masking systematically suppresses interfering gradient components and thus reduces gradient-norm variance. When combined with experience replay, gradient masking effectively neutralizes the instability introduced by replay. As a result, NBSP preserves the benefits of experience replay while mitigating its key drawback.

(2) Parameter similarity. We further evaluate the cosine similarity between network parameters before and after learning a new task, with results reported in Table 3. The ordering of parameter similarity is: NBSP > SAC + gradient masking > SAC + experience replay > vanilla SAC. Both

experience replay and gradient masking help constrain parameter drift, but through different mechanisms. Experience replay anchors parameters by reintroducing historical information, yet stale and task-mismatched samples still produce conflicting gradients that cause considerable displacement. Gradient masking regulates harmful gradient directions, but without explicit historical supervision it cannot fully prevent forgetting. As a result, neither mechanism alone is sufficient, experience replay still allows substantial drift, and gradient masking alone cannot preserve comprehensive past knowledge. When combined, however, experience replay supplies essential past-task information while gradient masking stabilizes the update directions. This complementary interaction keeps parameter changes within a range that preserves previously learned behaviors, explaining why NBSP achieves substantially better stability–plasticity performance than either mechanism used in isolation.

**Neuron identification method**. To evaluate the proposed goal-oriented neuron identification method, we compare it with three alternative strategies: (1) random neuron identification, (2) identifying neurons with activation magnitude (Jung et al., 2020), and (3) identifying neurons with weight magnitude (Dohare et al., 2021). As shown in Table 4, our goal-oriented method consistently outperforms the other three methods across all three metrics: ASR, FM, and FWT, which confirms that our method effectively identifies neurons critical for knowledge retention, ensuring better stability and plasticity in cycling sequential task learning. ***These findings validate the necessity of task-specific, goal-oriented neuron identification in enhancing balance between stability and plasticity.***

Table 4: Results of ablation study of neuron identification methods.

| Metrics | (window-open → window-close) | | | | (drawer-open → drawer-close) | | | | (button-press-topdown → window-open) | | | |
|---|---|---|---|---|---|---|---|---|---|---|---|---|
| | activation | weight | random | ours | activation | weight | random | ours | activation | weight | random | ours |
| ASR ↑ | 0.65±0.30 | 0.73±0.20 | 0.78±0.09 | **0.90±0.04** | 0.82±0.06 | 0.51±0.17 | 0.72±0.26 | **0.96±0.02** | 0.75±0.01 | 0.93±0.06 | 0.72±0.01 | **0.95±0.05** |
| FM ↓ | 0.56±0.37 | 0.44±0.31 | 0.42±0.13 | **0.18±0.01** | 0.44±0.16 | 0.67±0.00 | 0.41±0.28 | **0.07±0.06** | 0.65±0.02 | 0.15±0.12 | 0.70±0.05 | **0.08±0.12** |
| FWT ↑ | 0.73±0.35 | 0.81±0.22 | 0.90±0.06 | **0.96±0.02** | 0.98±0.02 | 0.69±0.22 | 0.83±0.23 | **0.98±0.01** | **0.99±0.00** | 0.98±0.02 | 0.96±0.02 | 0.98±0.01 |

**Actor and critic**. To get a deeper understanding of the individual roles of the actor and critic in DRL agents, we evaluate NBSP with that only applied on actor and critic. The results are shown in Table 5. ***The results indicate that both the actor and critic networks are essential for striking an optimal balance between stability and plasticity. Notably, the critic proves to be the more critical module in balancing this trade-off***, which aligns with the insight from Ma et al. (2024) that plasticity loss in the critic serves as the principal bottleneck impeding efficient training in DRL. ***We further investigate this phenomenon by examining the training dynamics of actor–critic RL methods and obtain three key observations***: (1) Actor updates are driven by critic feedback; thus, even when RL skill neurons in the actor are masked, the influence of critic can still lead to adaptation to new tasks at the cost of prior knowledge. (2) Applying NBSP to the critic indirectly constrains the actor. (3) The recursive updates of critic, with its target network maintained by an exponential moving average, help preserve previous knowledge while integrating new skills. These findings highlight the distinct roles of the actor and critic in balancing stability and plasticity, offering valuable guidance for future research.

Table 5: Results of ablation study of the actor and critic modules.

| Metric | (window-open → window-close) | | | (drawer-open → drawer-close) | | | (button-press-topdown → window-open) | | |
|---|---|---|---|---|---|---|---|---|---|
| | actor | critic | both | actor | critic | both | actor | critic | both |
| ASR ↑ | 0.76 ± 0.10 | 0.79 ± 0.05 | **0.90 ± 0.04** | 0.79 ± 0.05 | 0.86 ± 0.02 | **0.96 ± 0.02** | 0.81 ± 0.11 | 0.85 ± 0.16 | **0.95 ± 0.05** |
| FM ↓ | 0.58 ± 0.19 | 0.48 ± 0.09 | **0.18 ± 0.01** | 0.55 ± 0.15 | 0.31 ± 0.03 | **0.07 ± 0.06** | 0.45 ± 0.28 | 0.35 ± 0.38 | **0.08 ± 0.12** |
| FWT ↑ | **0.97 ± 0.04** | 0.94 ± 0.05 | 0.96 ± 0.02 | **0.99 ± 0.01** | 0.96 ± 0.02 | 0.98 ± 0.01 | 0.95 ± 0.01 | 0.95 ± 0.03 | **0.98 ± 0.01** |

**The proportion of RL skill neurons**. To evaluate the impact of the proportion of RL skill neurons $m$ on the performance of NBSP, we experiment with various proportions on the (button-press-topdown → window-open) cycling tasks. The results, shown in Figure 4, reveal an interesting trend: ***as the proportion of RL skill neurons increases, the ASR improves initially, but begins to decline after reaching a certain threshold***. Specifically, when the proportion of masked neurons is too small, not all neurons that encode task-specific skills can be correctly identified. As a result, the true RL skill neurons must adjust their

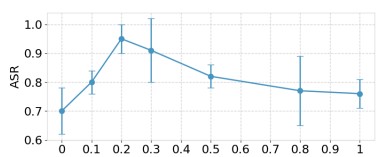

Figure 4: Performance of NBSP with different proportions of RL skill neurons.

activations to accommodate new tasks, ultimately reducing stability. Conversely, when the proportion becomes too large, neurons that do not encode task-specific skills may be mistakenly selected as

 Crucially, our experiments demonstrate that NBSP consistently delivers ASR improvements across a reasonably broad range of values (from 0.15 to 0.3). 

**More ablation results**. (1) The experience replay buffer stores information from all previously encountered tasks. (2) NBSP performs well within certain robust ranges of hyperparameters, making it easier to tune. (3) Task order does affect performance, but the impact is modest. (4) Vanilla PPO performs worse than vanilla SAC in our setting, and NBSP helps achieve a better balance. However, the effect is less pronounced than that of SAC. Please refer to Appendix C.10 for more details.

**The role of RL skill neurons**. Remarkably, we would like to highlight that RL skill neurons alone cannot resolve catastrophic forgetting, those non–RL skill neurons, influenced by the environment and context, also contribute to task success, and their forgetting can likewise cause failures. We believe neuron-level research for generalization across tasks in a more general sense would be an important future direction. Within our current framework, the primary contribution of RL skill neurons is mitigating catastrophic interference by preventing the overwriting the encoded knowledge during new task learning, which in turn synergizes with experience replay to alleviate catastrophic forgetting.

### 4.3 EXPERIMENT ON OTHER BENCHMARKS

Table 6: Results of NBSP with other baselines on the Atari benchmark.

| Cycling sequential games | Metrics | Methods | | | | | | | | | |
|---|---|---|---|---|---|---|---|---|---|---|---|
| | | EWC | NPC | ANCL | CoTASP | CRelu | CBP | PI | NE | UPGD | NBSP |
| (Pong → Bowling) | AR ↑ | $0.66 \pm 0.07$ | $0.51 \pm 0.02$ | $0.42 \pm 0.29$ | $-0.05 \pm 0.02$ | $0.02 \pm 0.00$ | $-0.09 \pm 0.00$ | $0.53 \pm 0.01$ | $0.45 \pm 0.03$ | $0.19 \pm 0.08$ | $\mathbf{0.87 \pm 0.01}$ |
| | FM ↓ | $0.58 \pm 0.20$ | $0.51 \pm 0.04$ | $0.46 \pm 0.31$ | $0.07 \pm 0.01$ | $\mathbf{0.01 \pm 0.00}$ | $0.06 \pm 0.00$ | $0.78 \pm 0.02$ | $0.66 \pm 0.04$ | $0.46 \pm 0.09$ | $\mathbf{0.05 \pm 0.03}$ |
| | FWT ↑ | $0.70 \pm 0.02$ | $0.35 \pm 0.02$ | $0.47 \pm 0.31$ | $-0.05 \pm 0.05$ | $0.02 \pm 0.01$ | $-0.09 \pm 0.00$ | $0.60 \pm 0.00$ | $0.47 \pm 0.02$ | $0.19 \pm 0.05$ | $\mathbf{0.72 \pm 0.01}$ |
| (BankHeist → Alien) | AR ↑ | $0.46 \pm 0.01$ | $0.38 \pm 0.06$ | $0.46 \pm 0.01$ | $-0.08 \pm 0.05$ | $0.08 \pm 0.05$ | $0.12 \pm 0.02$ | $0.48 \pm 0.14$ | $0.39 \pm 0.12$ | $0.28 \pm 0.01$ | $\mathbf{0.57 \pm 0.02}$ |
| | FM ↓ | $0.98 \pm 0.02$ | $0.46 \pm 0.14$ | $0.98 \pm 0.03$ | $\mathbf{0.27 \pm 0.04}$ | $0.52 \pm 0.29$ | $0.44 \pm 0.09$ | $0.88 \pm 0.27$ | $0.85 \pm 0.11$ | $0.60 \pm 0.03$ | $0.65 \pm 0.07$ |
| | FWT ↑ | $0.71 \pm 0.02$ | $0.37 \pm 0.03$ | $0.72 \pm 0.01$ | $-0.16 \pm 0.07$ | $0.28 \pm 0.11$ | $0.30 \pm 0.05$ | $\mathbf{0.73 \pm 0.26}$ | $0.63 \pm 0.09$ | $0.28 \pm 0.02$ | $0.72 \pm 0.05$ |

We further evaluate NBSP on the Atari and DMC benchmarks to assess its generalization ability. Atari games feature discrete action spaces and DMC is a widely recognized benchmark for continuous control tasks. Episode returns are used to evaluate the performance of each task and the results are presented in Table 6 and Table 7. On the Atari games, NBSP demonstrates superior performance in balancing stability and plasticity, outperforming other

Table 7: Results of NBSP on the DMC benchmark.

| Cycling sequential tasks | Metrics | Methods | |
|---|---|---|---|
| | | Vanilla SAC | NBSP |
| (Cartpole Swingup → Cartpole Balance) | AR ↑ | $746.80 \pm 5.26$ | $\mathbf{843.47 \pm 11.39}$ |
| | FM ↓ | $307.42 \pm 16.41$ | $\mathbf{59.26 \pm 10.89}$ |
| | FWT ↑ | $874.63 \pm 8.22$ | $\mathbf{883.32 \pm 6.69}$ |
| (Walker Walk → Walker Stand) | AR ↑ | $790.26 \pm 54.58$ | $\mathbf{861.09 \pm 24.99}$ |
| | FM ↓ | $272.26 \pm 67.62$ | $\mathbf{170.63 \pm 38.12}$ |
| | FWT ↑ | $899.44 \pm 26.46$ | $\mathbf{914.59 \pm 30.07}$ |

baselines across key evaluation metrics, including AR (Average Return), FM, and FWT, as with the Meta-World benchmark. NBSP also demonstrates its advantage compared to vanilla SAC on the DMC benchmark. These results further support that ***NBSP exhibits excellent generalization in balance stability and plasticity across the Meta-World, Atari and DMC benchmarks.***

## 5 CONCLUSION

This work addresses the fundamental issue of the stability-plasticity dilemma in DRL. To tackle this problem, we introduce the concept of RL skill neurons by identifying neurons that significantly contribute to knowledge retention, building upon which we then propose the Neuron-level Balance between Stability and Plasticity framework, by employing gradient masking and experience replay techniques on RL skill neurons. Experimental results on the Meta-World, Atari and DMC benchmarks demonstrate that NBSP significantly outperforms existing methods in managing the stability-plasticity trade-off. Future research could explore the application of RL skill neurons like model distillation and extend NBSP to other learning paradigms, such as supervised learning.

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

# A  RELATED WORK

**Balance between stability and plasticity**. In DRL, the agent faces a fundamental challenge: the stability-plasticity dilemma, first introduced by Carpenter & Grossberg (1988). Recent research has proposed various strategies to address this issue by balancing stability and plasticity.

Replay-based methods are widely employed to enhance stability by reusing experiences from past distributions. For example, Chaudhry et al. (2018b) introduced A-GEM, which combines episodic memory to ensure that the average loss of prior tasks does not increase when learning a new task. Similarly, Wolczyk et al. (2022) proposed ClonEx-SAC, which uses actor behavioral cloning and best-return exploration to boost performance in CRL. To reduce storage requirements, pseudo-rehearsals generated from a generative model have also been proposed (Atkinson et al., 2021a).

Maintaining the expressiveness of neurons is key to preserving plasticity. Nikishin et al. (2022b) proposed a mechanism that periodically resets a portion of the agent's network to counteract plasticity loss. Likewise, Nikishin et al. (2024) introduced plasticity injection, a lightweight intervention that enhances network plasticity without increasing trainable parameters or introducing prediction bias. The Reset & Distillation (R&D) framework combines resetting the online actor-critic network for new tasks with offline distillation of knowledge from previous action probabilities, effectively retaining plasticity (Ahn et al., 2024). Additionally, Abbas et al. (2023) proposed the Concatenated ReLUs (CReLUs) activation function to prevent activation collapse, thereby alleviating plasticity degradation.

Modularity-based approaches have shown promise in balancing stability and plasticity by decoupling task-specific and general knowledge. For instance, Anand & Precup (2024) decomposed the value function into a permanent value function, which captures persistent knowledge, and a transient value function, which facilitates rapid adaptation. Yang et al. (2020) designed a routing network to estimate task-specific routing strategies, reconfigure the base network, and combine routes using a soft modularity mechanism, making it effective for sequential tasks. Similarly, Mendez et al. (2022) proposed a compositional lifelong RL framework that uses accumulated neural components to accelerate learning for new tasks while preserving performance on past tasks via offline RL and replayed experiences.

**Neuron-level Research**. Recent research highlights that not all neurons remain active across varying contexts, and this neuron sparsity is often positively correlated with task-specific performance (Xu et al., 2024). Building on this insight, numerous studies have focused on identifying and leveraging skill neurons to interpret network behavior and tackle specific challenges, achieving significant advancements. For example, skill neurons in pre-trained Transformers, which demonstrate strong predictive value for task labels, have been utilized for network pruning to enhance efficiency and improve transferability (Wang et al., 2022). Sokar et al. (2023) investigate dormant neurons in deep reinforcement learning and propose a method to recycle them during training. Similarly, Dravid et al. (2023) introduce Rosetta Neurons, enabling cross-class alignments and transformations without specialized training. In large language models, language-specific neurons have been identified to control output languages by selective activation or deactivation (Tang et al., 2024), while safety neurons have been analyzed to enhance safety alignment through mechanistic interpretability (Chen et al., 2024).

Despite these achievements, the exploration of skill neurons in DRL remains limited. Existing neuron-level approaches primarily focus on task-specific sub-network selection. For instance, CoTASP learns hierarchical dictionaries and meta-policies to generate sparse prompts and extract sub-networks as task-specific policies (Yang et al., 2023). Similarly, Mallya & Lazebnik (2018) sequentially allocate multiple tasks within a single network through iterative pruning and re-training, balancing performance and storage efficiency. Unlike these methods, our work identifies RL skill neurons specifically tailored to deep reinforcement learning, ensuring a balance between stability and plasticity by preserving the task-relevant knowledge encoded in these neurons while allowing for fine-tuning.

# B  PRELIMINARY

## B.1  MARKOV DECISION PROCESS (MDP)

A Markov Decision Process(MDP) is a framework used to describe a problem involving learning from actions to achieve a goal. Almost all reinforcement learning problems can be characterized

as a Markov Decision Process. Each MDP is defined by a tuple $< S, A, P, R, \gamma >$, where $S$ and $A$ represent state and action spaces respectively. The transition dynamics of the MDP are defined by the function $P : S \times A \times S \rightarrow [0, 1]$, which represents the probability of transitioning from a give state $s$ with action $a$ to state $s'$. The reward function is represented by $R : S \times A \times S \rightarrow \mathbb{R}$, and $\gamma \in (0, 1)$ is the discount factor. At each time step $t$, an agent observes the state of the environment, denoted as $s_t$, and selects an action $a_t$ according to a policy $\pi(a|s)$. One time step later, the agent receives a numerical reward $r_{t+1}$ and transitions to a new state $s_{t+1}$. In the simplest case, the return is the sum of the rewards when the agent–environment interaction naturally breaks into subsequences, which we refer to episodes (Sutton, 2018).

## B.2 SOFT ACTOR-CRITIC (SAC)

Soft Actor-Critic (SAC) is an off-policy actor-critic deep reinforcement learning algorithm that leverages maximum entropy to promote exploration. This work employs SAC to train a policy that effectively balances stability and plasticity , chosen for its sample efficiency, excellent performance, and robust stability. In this framework, the actor aims to maximize both the expected reward and the entropy of the policy. The parameters $\phi$ of the actor are optimized by minimizing the following loss function:

$$J_\pi(\phi) = E_{s_t \sim D, a_t \sim \pi_\phi}[\alpha log \pi_\phi(a_t|s_t) - Q_\theta(s_t, a_t)],$$

where $D$ is the replay buffer, $\alpha$ is the temperature parameter controlling the trade-off between exploration and exploitation, $\theta$ denotes the parameters of the critic network, $\pi_\phi$ represents the policy learned by the actor $\phi$ , and $Q_\theta$ denotes the Q-value estimated by the critic $\theta$. The critic network is trained to minimize the squared residual error:

$$J_Q(\theta) = E_{(s_t, a_t, s_{t+1}) \sim D}[\frac{1}{2}(Q_\theta(s_t, a_t) - r_t - \gamma \hat{V}(s_{t+1})],$$

$$\hat{V}(s_t) = E_{a_t \sim \pi_\phi}[Q_\theta(s_t, a_t) - \alpha log \pi_\phi(a_t|s_t)],$$

where $\gamma$ represents the discount factor.

## B.3 NEURON

In neural networks, various components, such as blocks and layers, play distinct roles. Here, we define a neuron as a single output dimension from a layer. For example, in a fully connected layer, each output dimension corresponds to a neuron. Similarly, in a convolutional layer, each output channel represents a neuron. Furthermore, following the terminology used by Sajjad et al. (2022), we classify neurons that encapsulate a single concept as focused neurons, while a group of neurons collectively representing a concept are termed group neurons.

## C EXPERIMENT

### C.1 BASELINE

**EWC**: Elastic Weight Consolidation (EWC) (Kirkpatrick et al., 2017) addresses the challenge of catastrophic forgetting by allowing neural networks to retain proficiency in previously learned tasks even after a long hiatus. It achieves this by selectively slowing down learning for weights that are crucial for retaining knowledge of these tasks. This approach has demonstrated excellent performance in sequentially solving a series of classification tasks, such as those in the MNIST handwritten digit dataset, and in learning several Atari 2600 games sequentially.

**NPC**: Neuron-level Plasticity Control (NPC) (Paik et al., 2019) preserves the existing knowledge from the previous tasks by controlling the plasticity of the network at the neuron level. NPC estimates the importance value of each neuron and consolidates important neurons by applying lower learning rates, rather than restricting individual connection weights to stay close to the values optimized for the previous tasks. The experimental results on the several classification datasets show that neuron-level consolidation is substantially effective.

**ANCL**: Auxiliary Network Continual Learning (ANCL) is an innovative approach that incorporates an auxiliary network to enhance plasticity within a model that primarily emphasizes stability. Specifically,

this framework introduces a regularizer that effectively balances plasticity and stability, achieving superior performance over strong baselines in both task-incremental and class-incremental learning scenarios.

**CoTASP**: Continual Task Allocation via Sparse Prompting (CoTASP) (Yang et al., 2023) learns over-complete dictionaries to produce sparse masks as prompts extracting a sub-network for each task from a meta-policy network. Hence, relevant tasks share more neurons in the meta-policy network due to similar prompts while cross-task interference causing forgetting is effectively restrained. It outperforms existing continual and multi-task RL methods on all seen tasks, forgetting reduction, and generalization to unseen tasks. **CoTASP**: Continual Task Allocation via Sparse Prompting (CoTASP) (Yang et al., 2023) learns over-complete dictionaries to produce sparse masks as prompts extracting a sub-network for each task from a meta-policy network. Hence, relevant tasks share more neurons in the meta-policy network due to similar prompts while cross-task interference causing forgetting is effectively restrained. It outperforms existing continual and multi-task RL methods on all seen tasks, forgetting reduction, and generalization to unseen tasks.

**CRelu**: Concatenated ReLUs (CReLUs) (Abbas et al., 2023) is a simple activation function that concatenates the input with its negation and applies ReLU to the result. It performs effectively in facilitating continual learning in a changing environment.

**CBP**: Continual BackPropagation (CBP) (Dohare et al., 2024) reinitializes a small number of units during training, typically fewer than one per step. To prevent disruption of what the network has already learned, only the least-used units are considered for reinitialization. It shows great performance on Continual ImageNet and class-incremental CIFAR-100.

**PI**: Plasticity Injection (PI) (Nikishin et al., 2024) freeze the parameters $\theta$ and introduce a new set of parameters $\theta\prime$ sampled from random initialization at some point in training, where the network might have started losing plasticity. The results on Atari show that plasticity injection attains stronger performance compared to alternative methods while being computationally efficient.

**NE**: Neuroplastic Expansion(NE) (Liu et al., 2024) maintains learnability and adaptability throughout the entire training process by dynamically growing the network from a smaller initial size to its full dimension.

**UPGD**: Utility-based Perturbed Gradient Descent (UPGD) (Elsayed & Mahmood, 2024) combines gradient updates with perturbations, where it applies smaller modifications to more useful units, protecting them from forgetting, and larger modifications to less useful units, rejuvenating their plasticity.

## C.2 BENCHMARK

**Meta-World**. Meta-World is an open-source benchmark for meta-reinforcement learning and multitask learning, comprising 50 distinct robotic manipulation tasks (Yu et al., 2020).

All tasks are executed by a simulated Sawyer robot, with the action space defined as a 2-tuple: the change in the 3D position of the end-effector, followed by a normalized torque applied to the gripper fingers.

The observation space has a consistent dimensionality of 39, although different dimensions correspond to various aspects of each task. Typically, the observation space is represented as a 6-tuple, including the 3D Cartesian position of the end-effector, a normalized measure of the gripper's openness, the 3D position and the quaternion of the first object, the 3D position and quaternion of the second object, all previous measurements within the environment, and the 3D position of the goal.

The reward function for all tasks is structured and multi-component, aiding in effective policy learning for each task component. With this design, the reward functions maintain a similar magnitudes across tasks, generally ranging between 0 and 10. The descriptions of the six tasks used in our experiments are listed below, and the appearance of these tasks is shown in Figure 5.

- **drawer-open**: Open a drawer, with randomized drawer positions.
- **drawer-close**: Push and close a drawer, with randomized drawer positions.
- **window-open**: Push and open a window, with randomized window positions.

- **window-close**: Push and close a window, with randomized window positions.
- **door-open**: Open a door with a revolving joint. Randomize door positions.
- **button-press-topdown**: Press a button from the top. Randomize button positions.

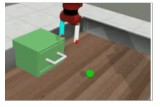 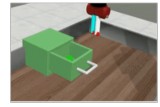 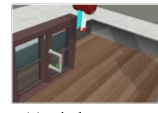 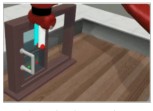 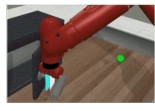 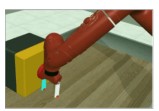

(a) drawer-open     (b) drawer-close     (c) window-open     (d) window-close     (e) door-open     (f) button-press-topdown

Figure 5: Tasks in the Meta-World benchmark used in our experiments.

**Atari**. Atari environments are simulated using the Arcade Learning Environment (ALE) (Bellemare et al., 2013) via the Stella emulator.

Each environment utilizes a subset of the full action space, which includes actions like NOOP, FIRE, UP, RIGHT, LEFT, DOWN, UPRIGHT, UPLEFT, DOWNRIGHT, DOWNLEFT, UPFIRE, RIGHTFIRE, LEFTFIRE, DOWNFIRE, UPRIGHTFIRE, UPLEFTFIRE, DOWNRIGHTFIRE, and DOWNLEFTFIRE. By default, most environments employ only a smaller subset of these actions, excluding those that have no effect on gameplay.

Observations in Atari environments are RGB images displayed to human players, with $obs\_type =$ "$rgb$", corresponding to an observation space defined as $Box(0, 255, (210, 160, 3), np.uint8)$.

The specific reward dynamics vary depending on the environment and are typically detailed in the game's manual.

The descriptions of the four games used in our experiments are listed below (Foundation, 2024), and the appearance of these games is shown in Figure 6.

- **Bowling**: The goal is to score as many points as possible in a 10-frame game. Each frame allows up to two tries. Knocking down all pins on the first try is called a "strike", while doing so on the second try is a "spare". Failing to knock down all pins in two attempts results in an "open" frame.
- **Pong**: You control the right paddle and compete against the computer-controlled left paddle. The objective is to deflect the ball away from your goal and into the opponent's goal.
- **BankHeist**: You play as a bank robber trying to rob as many banks as possible while avoiding the police in maze-like cities. You can destroy police cars using dynamite and refill your gas tank by entering new cities. Lives are lost if you run out of gas, are caught by the police, or run over your own dynamite.
- **Alien**: You are trapped in a maze-like spaceship with three aliens. Your goal is to destroy their eggs scattered throughout the ship while avoiding the aliens. You have a flamethrower to fend them off and can occasionally collect a power-up (pulsar) that temporarily enables you to kill aliens.

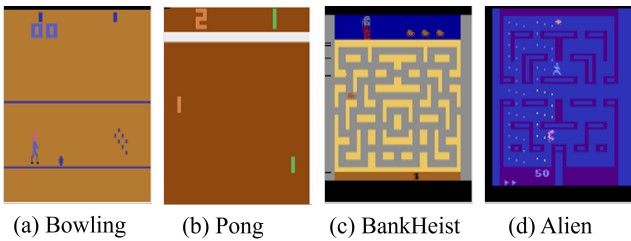

(a) Bowling     (b) Pong     (c) BankHeist     (d) Alien

Figure 6: Games in the Atari benchmark used in our experiments.

**DMC**. DeepMind Control Suite (DMC) (Tunyasuvunakool et al., 2020) is a widely used set of standardized environments for reinforcement learning research, designed to provide a challenging

benchmark for continuous control tasks. Built upon the powerful MuJoCo physics engine, it offers a diverse collection of simulations, ranging from simple cart-pole balancing to complex locomotion tasks for cheetah-like quadrupeds and humanoids.

## C.3 EXPERIMENT SETTING

For all experiments, we utilize the open-source PyTorch implementation of Soft Actor-Critic (SAC) provided by CleanRL (Huang et al., 2022) on a single RTX2080Ti GPU. CleanRL is a Deep Reinforcement Learning library that offers high-quality, single-file implementations with research-friendly features. The code is both clean and straightforward, and we adhere to the configurations provided by CleanRL. During training, we employ an $\epsilon$-greedy exploration policy at the start, setting $\epsilon = 1$ for the first $10^4$ time steps to promote exploration. The environment is wrapped using Gym wrappers to facilitate experimentation. For the Meta-World benchmark, we utilize the RecordEpisodeStatistics wrapper to gather episode statistics. For the Atari benchmark, in addition to RecordEpisodeStatistics, we preprocess the $210 \times 160$ pixel images by downsampling them to $84 \times 84$ using bilinear interpolation, converting the RGB images to the YUV format, and using only the grayscale channel. Additionally, we set a maximum limit on the number of noop and skip steps to standardize the exploration.

Regarding network architecture, we use the same actor and critic networks for all tasks within the same benchmark to ensure consistency. For the Meta-World benchmark, we employ a neural network comprising four fully connected layers, of which the hidden size is [768, 768, 768]. For the Atari benchmark, we use a convolutional neural network (CNN) with three convolutional layers featuring 32, 64, and 64 channels, respectively, followed by three fully connected layers, of which the hidden size is [768, 768].

To reduce randomness and enhance the reliability of our results, we train each agent using three random seeds. Additional hyper-parameters for the SAC algorithm applied in the Meta-World, Atari and DMC benchmarks are detailed in Table 8.

Table 8: Hyper-parameters of SAC in our experiments.

| Parameters | Values for Meta-World | Values for Atari | Values for DMC |
|---|---|---|---|
| Initial collect steps | 10000 | 20000 | 10000 |
| Discount factor | 0.99 | 0.99 | 0.99 |
| Training environment steps | $10^6$ | $1.5 \times 10^6, 3 \times 10^6$ | $3 \times 10^5$ |
| Testing environment steps | $10^5$ | $10^5$ | $10^5$ |
| Replay buffer size | $10^6$ | $2 \times 10^5$ | $10^6$ |
| Updates per environment step (Replay Ratio) | 2 | 4 | 2 |
| Target network update period | 1 | 8000 | 1 |
| Target smoothing coefficient | 0.005 | 1 | 0.005 |
| Optimizer | Adam | Adam | Adam |
| Policy learning rate | $3 \times 10^{-4}$ | $10^{-4}$ | $3 \times 10^{-4}$ |
| Q-value learning rate | $10^{-3}$ | $10^{-4}$ | $10^{-3}$ |
| Minibatch size | 256 | 64 | 256 |
| Alpha | 0.2 | 0.2 | 0.2 |
| Autotune | True | True | True |
| Average environment steps of success rate | 10 | - | - |
| Stable threshold to finish training | 0.9 | - | - |
| Replay interval | 10 | 10 | 10 |
| No-op max | - | 30 | - |
| Target entropy scale | - | 0.89 | - |
| Storing experience size | $10^5$ | $10^5$ | $10^5$ |
| Average steps | $5 \times 10^4$ | $5 \times 10^4$ | $5 \times 10^4$ |
| Proportion of RL skill neurons | 0.2 | 0.2 | 0.2 |

## C.4 METRICS

Overall performance is commonly assessed using the **Average Success Rate (ASR)**, analogous to the AIA metric (Wang et al., 2024). Let $sr_{i,j}$ represent the success rate on the $j$-th task after completing the learning of the $i$-th task ($i \geq j$), $H$ denote the number of tasks. The ASR is defined as follows.

The higher the ASR, the better the method balances stability and plasticity.

$$ASR = \frac{1}{H} \sum_{i=1}^{H} \frac{1}{i} \sum_{i \geq j} sr_{i,j}, \tag{9}$$

To evaluate the stability of the agent, we utilize the **Forgetting Measure (FM)** (Chaudhry et al., 2018a). The lower the FM, the better the method maintains stability, which is calculated as:

$$FM = \frac{1}{H-1} \sum_{i=2}^{H} \frac{1}{i-1} \sum_{i \geq j} \max_{l \in \{1,...,i-1\}} (sr_{l,j} - sr_{i,j}). \tag{10}$$

To assess the plasticity of the agent, we employ the **Forward Transfer (FWT)** metric (Lopez-Paz & Ranzato, 2017), which is calculated as follows:

$$FWT = \frac{1}{H} \sum_{i=1}^{H} sr_{i,i}. \tag{11}$$

The higher the FWT, the better the method maintains plasticity.

For the Meta-World benchmark, the average success rate is computed over 20 episodes. For the Atari benchmark, the success rate is replaced by the return of each episode. We normalize the return for each game to obtain summary statistics across games, as follows:

$$R = \frac{r_{agent} - r_{random}}{r_{human} - r_{random}}, \tag{12}$$

where $r_{agent}$ represents the average return evaluated over $10^5$ steps, the random score $r_{random}$ and human score $r_{human}$ are consistent with those used by Mnih et al. (2015), as detailed in Table 9.

Table 9: Normalization scores of Atari games.

| games | $r_{random}$ | $r_{human}$ |
|---|---|---|
| Bowling | 23.1 | 154.8 |
| Pong | -20.7 | 9.3 |
| BankHeist | 14.2 | 734.4 |
| Alien | 227.5 | 6875 |

For the Atari benchmark tasks, the overall performance is evaluated by Average Return (AR), which is analogous to ASR in the Meta-World benchmark. It is calculated as follows:

$$AR = \frac{1}{k} \sum_{i=1}^{k} \frac{1}{i} \sum_{i \geq j} R_{i,j}, \tag{13}$$

where $R_{i,j}$ represents the average return evaluated on the $j$-th task after completing the learning of the $i$-th task ($i \geq j$), and $k$ represents the number of tasks. A higher AR indicates better performance in balancing stability and plasticity.

## C.5 RL SKILL NEURONS

To validate the existence of RL skill neurons in sequential task learning instead of single task learning, we conduct an additional analysis comparing the activation distributions of neurons when learning button-press-topdown in isolation versus learning button-press-topdown and window-open simultaneously. As shown in Figure 7, the activation distribution of a representative neuron remains highly correlated with task success, regardless of whether it is learned in isolation or alongside another skill. This observation supports our hypothesis that skill-specific neurons retain their essential role even in a sequential task learning scenario.

Additionally, we dig deeper into the identified RL skill neurons and separate them into general and specific skills. How to deeply investigate general skills is key for our future research. To explore

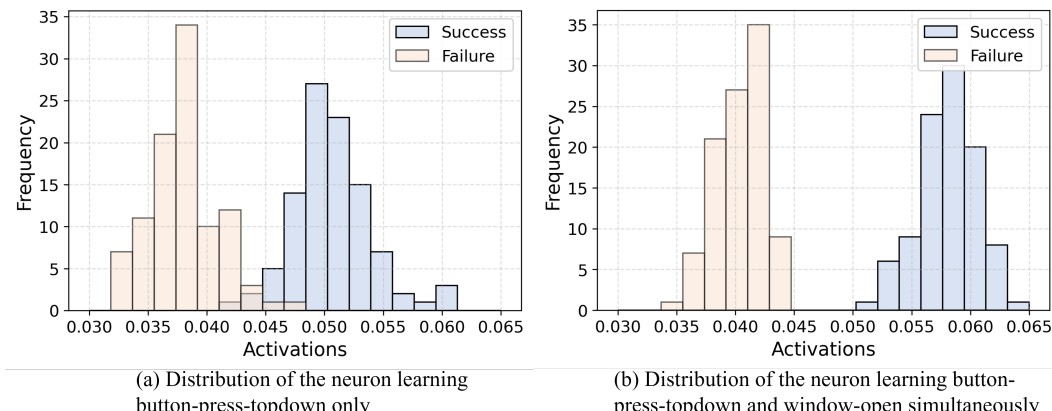

(a) Distribution of the neuron learning button-press-topdown only

(b) Distribution of the neuron learning button-press-topdown and window-open simultaneously

Figure 7: Distribution histogram of the activations of a neuron in two learning settings.

this, we design an experiment to verify the existence of general and specific skills. After sequentially training on the button-press-topdown and window-open tasks, we identify the RL skill neurons associated with each task. We hypothesize that the intersection set represents general skill neurons, while the difference set represents specific skill neurons. To validate this hypothesis, we zero out the outputs of these neurons separately. The results in Table 10 show that when the outputs of the general skill neurons are zeroed out, the agent fails to complete both tasks. In contrast, when the outputs of task-specific neurons are zeroed out, the agent can't complete the corresponding task but is still able to complete the other task. This confirms the existence of both general and specific skills.

Table 10: Results of zeroing out the output of general of specific skill neurons.

| tasks | zero out the intersection set | zero out the difference set of button-press-topdown relative to window-open | zero out the difference set of window-open relative to button-press-topdown |
|---|---|---|---|
| button-press-topdown | 0 | 0.33 | 1.00 |
| window-open | 0 | 1.0 | 0.42 |

## C.6 RESULTS OF VANILLA SAC

To validate the effectiveness of NBSP, it is essential to first confirm whether the vanilla SAC algorithm can successfully solve each task individually. So we conducted experiments by training a vanilla SAC agent on all tasks in our experiment. The results, presented in Figure 8, demonstrate that the vanilla SAC algorithm successfully learns all tasks in our experiment. This confirms that the balance between stability and plasticity is not an artifact of modifications to the SAC algorithm itself but rather a result of NBSP. Furthermore, the failure of other methods is not due to limitations of the SAC algorithm.

## C.7 RESULTS ON THE META-WORLD BENCHMARK

### C.7.1 RESULTS OF LONGER TASK SEQUENCE

Table 11: Results of ten task sequences on the Meta-world benchmark.

| | ASR ↑ | FM ↓ | FWT ↑ |
|---|---|---|---|
| vanilla SAC | $0.27 \pm 0.05$ | $0.79 \pm 0.07$ | $0.52 \pm 0.19$ |
| NE | $0.58 \pm 0.05$ | $0.44 \pm 0.04$ | $0.64 \pm 0.03$ |
| NBSP | $0.66 \pm 0.02$ | $0.32 \pm 0.06$ | $0.74 \pm 0.01$ |

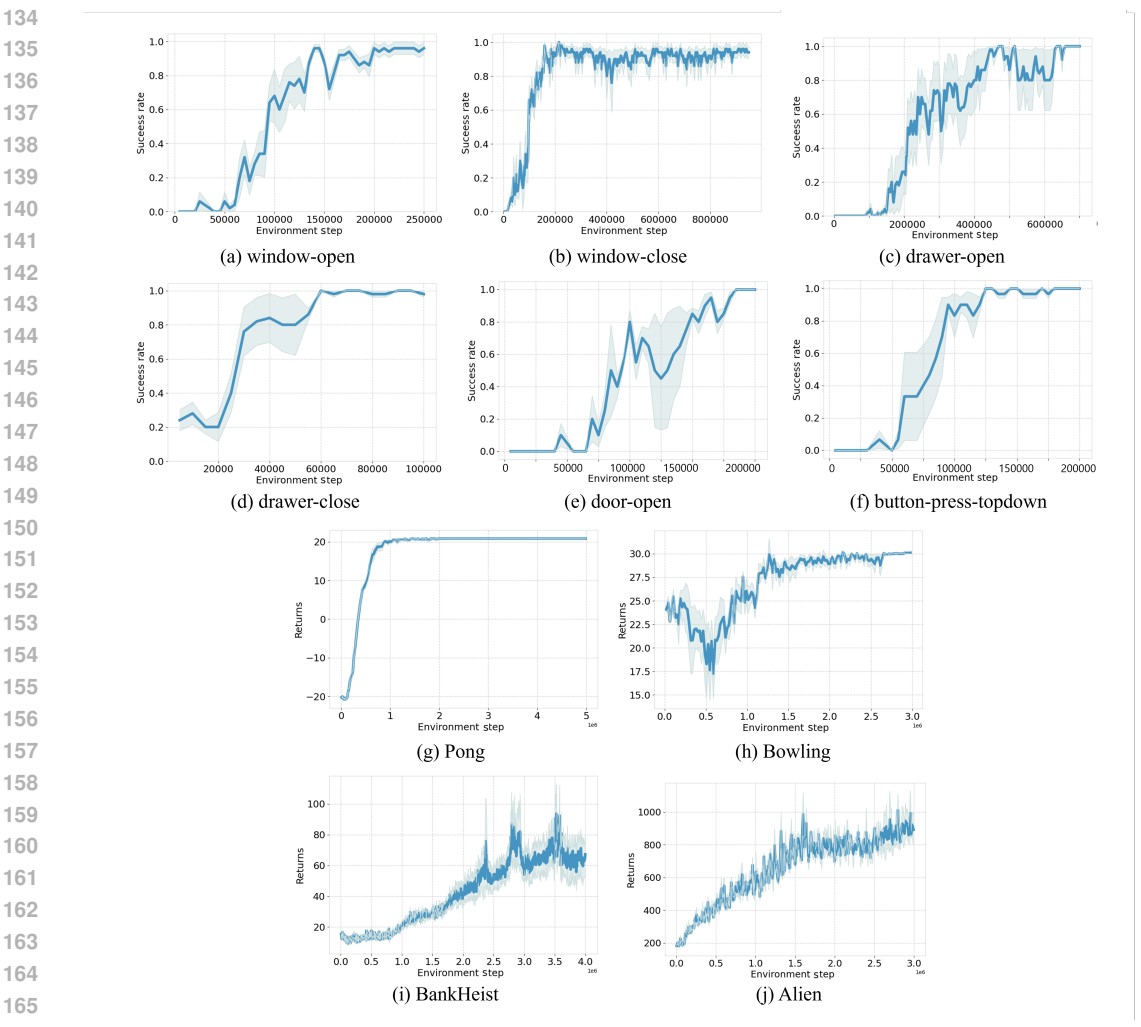

Figure 8: Training process of vanilla SAC on each individual task in our experiment.

To further evaluate NBSP under longer task sequences, we additionally conduct experiments on a 10-task sequence (button-press-topdown → window-close → door-open → drawer-close → drawer-open → door-close → button-press-topdown-wall → window-open → push → reach) in the Meta-World benchmark. As shown in Table 11, NBSP continues to outperform vanilla SAC by a large margin, particularly in terms of stability, where the forgetting metric decreases from 0.79 to 0.32. We also compare NBSP with NE, which is the strongest baseline in our earlier Meta-World experiments. Although NE achieves a better stability–plasticity balance than vanilla SAC, it still lags behind NBSP on the longer 10-task sequence. These results demonstrate that NBSP generalizes well beyond short task chains and is effective even in long-horizon task settings.

### C.7.2 LEARNING CURVE

The training process of the other four-tasks cycling task is shown in Figure 9, and those of the two-task cycling tasks are shown in Figure 10, Figure 11 and Figure 12 respectively. The same as found in Section 4.1, during the second cycle of learning the same task, the agent is able to master the task more rapidly.

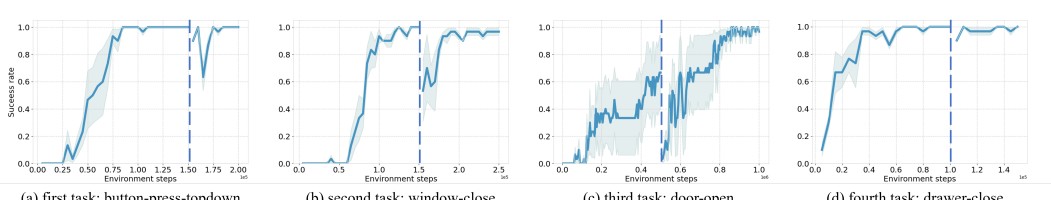

Figure 9: Training process of NBSP on (button-press-topdown $\rightarrow$ window-close $\rightarrow$ door-open $\rightarrow$ drawer-close) cycling task.

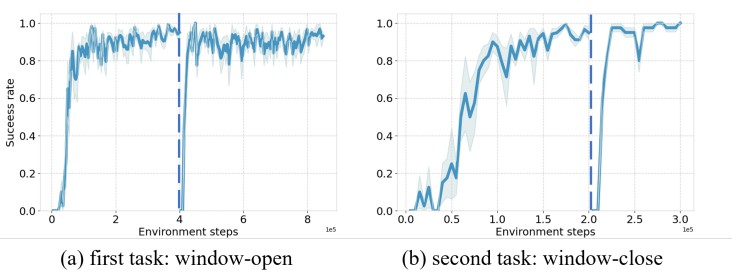

Figure 10: Training process of NBSP on (window-open $\rightarrow$ window-close) cycling task.

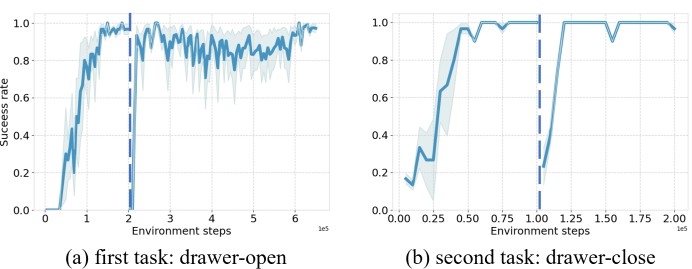

Figure 11: Training process of NBSP on (drawer-open $\rightarrow$ drawer-close) cycling task.

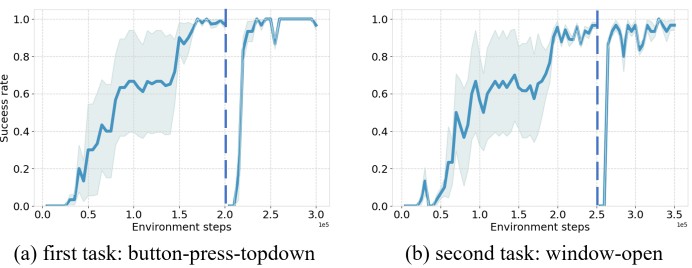

Figure 12: Training process of NBSP on (button-press-topdown $\rightarrow$ window-open) cycling task.

## C.8 RESULTS ON THE ATARI BENCHMARK

The training process of the two-task cycling tasks are shown in Figure 13, Figure 14.

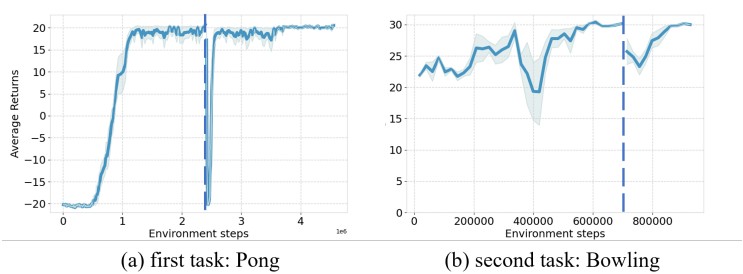

(a) first task: Pong          (b) second task: Bowling

Figure 13: Training process of NBSP on (Pong → Bowling) cycling task.

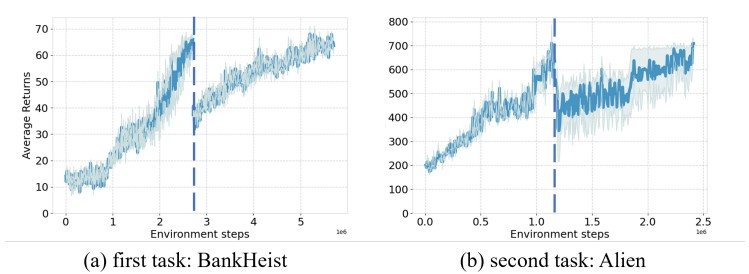

(a) first task: BankHeist          (b) second task: Alien

Figure 14: Training process of NBSP on (BankHeist → Alien) cycling task.

## C.9 RESULTS ON THE DMC BENCHMARK

The training process of the two-task cycling tasks are shown in Figure 15, Figure 16.

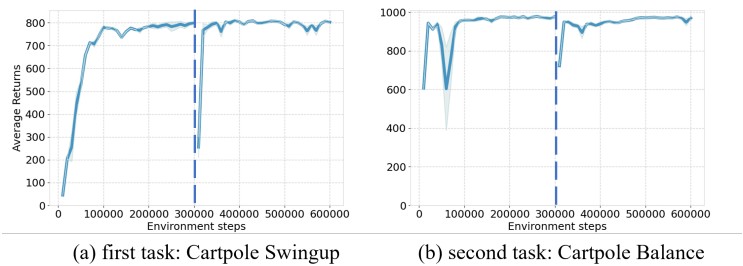

(a) first task: Cartpole Swingup          (b) second task: Cartpole Balance

Figure 15: Training process of NBSP on (Cartpole Swingup → Cartpole Balance) cycling task.

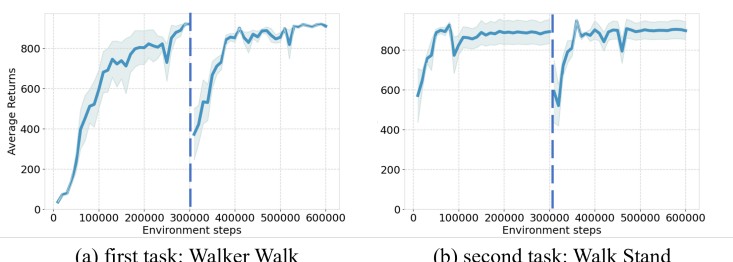

(a) first task: Walker Walk
(b) second task: Walk Stand

Figure 16: Training process of NBSP on (Walker Walker $\rightarrow$ Walker Stand) cycling task.

## C.10 ABLATION STUDY

### C.10.1 GRADIENT MASKING AND EXPERIENCE REPLAY

The results of the ablation study on two critical components, gradient masking and experience replay techniques, are shown in Table 12 for the (window-open $\rightarrow$ window-close) cycling task and in Table 13 for the (drawer-open $\rightarrow$ drawer-close) cycling task. From these results, it is evident that both gradient masking and experience replay techniques independently contribute to improving the stability of the agent while maintaining great plasticity. Furthermore, combining both techniques yields superior performance, demonstrating the enhanced effectiveness of their integration.

Table 12: Results of ablation study of gradient masking and experience replay techniques on (window-open $\rightarrow$ window-close) cycling task.

| Metrics | (button-press-topdown $\rightarrow$ window-open) | | | | |
| | vanilla SAC | only experience replay | only gradient masking | NBSP with hard gradient masking | NBSP |
|---|---|---|---|---|---|
| ASR $\uparrow$ | $0.63 \pm 0.02$ | $0.81 \pm 0.08$ | $0.78 \pm 0.11$ | $0.71 \pm 0.04$ | $\mathbf{0.90 \pm 0.04}$ |
| FM $\downarrow$ | $0.91 \pm 0.10$ | $0.41 \pm 0.13$ | $0.54 \pm 0.26$ | $0.54 \pm 0.13$ | $\mathbf{0.18 \pm 0.01}$ |
| FWT $\uparrow$ | $0.97 \pm 0.02$ | $0.96 \pm 0.01$ | $\mathbf{0.98 \pm 0.01}$ | $0.91 \pm 0.05$ | $0.96 \pm 0.02$ |

Table 13: Results of ablation study of gradient masking and experience replay techniques on (drawer-open $\rightarrow$ drawer-close) cycling task.

| Metrics | (button-press-topdown $\rightarrow$ window-open) | | | | |
| | vanilla SAC | only experience replay | only gradient masking | NBSP with hard gradient masking | NBSP |
|---|---|---|---|---|---|
| ASR $\uparrow$ | $0.67 \pm 0.05$ | $0.78 \pm 0.04$ | $0.74 \pm 0.01$ | $0.59 \pm 0.16$ | $\mathbf{0.96 \pm 0.02}$ |
| FM $\downarrow$ | $0.78 \pm 0.10$ | $0.48 \pm 0.10$ | $0.64 \pm 0.01$ | $0.52 \pm 0.35$ | $\mathbf{0.07 \pm 0.06}$ |
| FWT $\uparrow$ | $0.94 \pm 0.04$ | $0.97 \pm 0.01$ | $\mathbf{0.98 \pm 0.02}$ | $0.82 \pm 0.21$ | $\mathbf{0.98 \pm 0.01}$ |

### C.10.2 GRADIENT MASKING

To verify that the effectiveness of adaptive gradient masking goes beyond simply reducing the learning rate, we additionally conduct experiments on the cyclic task sequence (button-press-topdown $\rightarrow$ window-open) comparing three settings: (1) Lowering the learning rate only on RL skill neurons, (2) Lowering the learning rate on the entire network, (3) Our proposed NBSP gradient masking. The results are presented in Table 14. We observe the following:

- NBSP achieves the best balance between stability and plasticity.

  NBSP adaptively applies different degrees of masking to RL skill neurons based on their score. This allows high-score neurons to be more strongly preserved while still enabling low-importance neurons to tune. A constant scaled learning rate cannot capture this differentiation and therefore underperforms NBSP.

- Lowering the learning rate on RL skill neurons only offers partial benefits but remains inferior.

Treating all RL skill neurons uniformly ignores the heterogeneity in how different neurons contribute to task-specific behaviors. While this reduces interference to some extent, it still results in suboptimal stability–plasticity trade-offs.

• Lowering the learning rate across the entire network performs the worst.
  This merely slows down parameter updates without specifically protecting the neurons that encode task skills. As a result, RL skill neurons continue to be overwritten by new tasks, leading to high forgetting (FM) and degraded performance.

Overall, these comparisons show that the benefit of NBSP cannot be replicated by globally reducing the learning rate. The key advantage comes from adaptive and selective protection of RL skill neurons.

Table 14: Results of gradient masking and lowering the learning rate.

|  | ASR ↑ | FM ↓ | FWT ↑ |
|---|---|---|---|
| Lower the lr of RL Skill neuron | $0.86 \pm 0.07$ | $0.27 \pm 0.15$ | $0.97 \pm 0.02$ |
| Lower the lr of the entire network | $0.71 \pm 0.12$ | $0.68 \pm 0.34$ | $0.98 \pm 0.01$ |
| NBSP | $0.95 \pm 0.05$ | $0.08 \pm 0.12$ | $0.98 \pm 0.01$ |

### C.10.3 EXPERIENCE REPLAY

To investigate whether the experience replay buffer stores information from all previously encountered tasks, we conduct experiments on the cycling task sequence (button-press-topdown → window-close → door-open → drawer-close), where we compared two buffer configurations: (1) Storing only the most recent task's experience. (2) Storing experience from all past tasks. As Table 15 below clearly shows, restricting the buffer to just the previous task leads to significantly higher Forgetting Metric (FM) values, indicating greater forgetting and reduced stability. In comparison, storing data from all previously encountered tasks consistently improves stability.

Table 15: Performance under different replay buffer configurations.

| Buffer configurations | ASR | FM | FWT |
|---|---|---|---|
| Storing only the most recent task's experience | $0.51 \pm 0.12$ | $0.75 \pm 0.12$ | $0.90 \pm 0.20$ |
| Storing experience from all past tasks. | $0.74 \pm 0.07$ | $0.34 \pm 0.15$ | $0.95 \pm 0.06$ |

To further confirm this, we evaluate the success rates of the first three tasks after training on the fourth task, specifically under the setting where the experience replay buffer contained only data from the most recent task. Results in Table 16 show that when the agent is trained on the four task with the buffer restricted to experience from only the third task, it retained its ability to perform the third task but entirely failed on the first and second one. This validates that the experience replay buffer must store information from all previously encountered tasks. Omitting earlier tasks from the buffer directly leads to catastrophic forgetting.

Table 16: Performance of the first three tasks after training on the fourth task.

| Task | First task: button-press-topdown | Second task: window-close | Third task: door-open |
|---|---|---|---|
| Success rate | $0.00 \pm 0.00$ | $0.03 \pm 0.05$ | $0.80 \pm 0.08$ |

The results also demonstrate that RL skill neurons cannot resolve catastrophic forgetting on their own. For example, for the success of a specific task, those non-RL skill neurons could also matter, which may relate to factors such as the environments and contexts. And catastrophic forgetting of these non-RL skill cabilities could also lead to task failures. Under the current framework, the primary contribution of RL skill neurons is mitigating catastrophic interference by preventing overwriting the knowledge encoded in these neurons when learning new tasks. We would like to highlight that mitigating catastrophic interference could also benefit catastrophic forgetting, where RL skill neurons and experience replay buffer work in a synergic way to alleviate forgetting.

### C.10.4 BASELINES WITH EXPERIENCE REPLAY

To ensure fair comparison, we add experience replay to ANCL and CoTASP on the cycling task sequence (button-press-topdown → window-open). As shown in Table 17, ANCL with experience replay heavily improves stability (FM from 0.95 to 0.42), while CoTASP with experience replay gets no substantial performance gain, for the characteristics of tasks differ from the knowledge distribution captured by the existing dictionary, preventing the generation of effective sparse prompts. And NBSP achieves better performance on all metrics.

Table 17: Performance comparison of methods with experience replay.

| Method | ASR ↑ | FM ↓ | FWT ↑ |
|---|---|---|---|
| ANCL | $0.61 \pm 0.01$ | $0.95 \pm 0.05$ | $0.95 \pm 0.03$ |
| ANCL + experience replay | $0.83 \pm 0.07$ | $0.42 \pm 0.18$ | $0.97 \pm 0.02$ |
| CoTASP | $0.03 \pm 0.00$ | $0.01 \pm 0.00$ | $0.04 \pm 0.01$ |
| CoTASP + experience replay | $0.03 \pm 0.01$ | $0.01 \pm 0.01$ | $0.05 \pm 0.01$ |
| NBSP | $0.95 \pm 0.05$ | $0.08 \pm 0.12$ | $0.98 \pm 0.01$ |

### C.10.5 NEURONS IN THE LAST LAYER

We exclude the final layer from neuron scoring for two main reasons: (1) The final layer contains very few neurons (e.g., a single output unit in the critic). Masking such neurons would excessively constrain the network and directly damage its optimization capacity. (2) Final-layer activations correspond directly to network outputs. These neurons inherently play a disproportionately large role in determining performance. As a result, in practice they are more likely to be identified as RL Skill neuron. Thus masking them is more likely to degrade the learning dynamics.

To validate this design choice, we conducted additional experiments where the final layer was included in the neuron identification and masking process. The results shown in the Table 18 confirm our hypothesis. Overall performance deteriorates, primarily due to a substantial decrease in ASR, and plasticity is harmed (lower FWT), as restricting output-layer neurons prevents the network from efficiently adapting to new tasks, and stability also worsens, because once the final-layer neurons are masked, the network compensates by over-adjusting earlier RL skill neurons, inadvertently damaging previously acquired knowledge.

Table 18: Results of RL Skill neurons with/without the last layer neurons.

| | ASR ↑ | FM ↓ | FWT ↑ |
|---|---|---|---|
| include the last layer | $0.76 \pm 0.01$ | $0.41 \pm 0.02$ | $0.87 \pm 0.02$ |
| exclude the last layer | $0.95 \pm 0.05$ | $0.08 \pm 0.12$ | $0.98 \pm 0.01$ |

### C.10.6 SENSITIVITY ANALYSIS

We investigate the sensitivity of these parameters on a cycling task sequence (button-press-topdown → window-open).

**m**: As discussed in 4.2, the performance improves as $m$ increases, but it begins to decline after a certain threshold. The results in Figure 4 demonstrate that NBSP consistently delivers significant ASR improvements across a broad range of $m$ values (from 0.15 to 0.3), indicating that its selection is not overly sensitive within a practical operational range. Setting $m$ within this range yields strong performance gains across different tasks and benchmarks in our experiments.

$\alpha$: We vary $\alpha$ from 0.1 to 1.0, whose results are shown in Tabel 19. As $\alpha$ decreases, FM consistently improves, indicating better stability, while FWT remains relatively stable, suggesting that plasticity is not highly sensitive to $\alpha$. Notably, ASR performance is strong as long as $\alpha < 0.3$, Overall, NBSP is robust to the choice of $\alpha$ within this range.

Table 19: Effect of the hyperparameter $\alpha$ on performance.

| $\alpha$ | ASR ↑ | FM ↓ | FWT ↑ |
|---|---|---|---|
| 0.1 | $0.93 \pm 0.04$ | $0.07 \pm 0.10$ | $0.96 \pm 0.01$ |
| 0.2 | $0.95 \pm 0.05$ | $0.08 \pm 0.12$ | $0.98 \pm 0.01$ |
| 0.3 | $0.91 \pm 0.07$ | $0.13 \pm 0.16$ | $0.98 \pm 0.01$ |
| 0.5 | $0.85 \pm 0.01$ | $0.33 \pm 0.00$ | $0.98 \pm 0.01$ |
| 1.0 | $0.81 \pm 0.06$ | $0.48 \pm 0.15$ | $0.98 \pm 0.01$ |

$|\mathbf{D_{pre}}|$: We vary buffer sizes ranging from $1e2$ to $1e6$. The results are shown in Table 20, when the buffer size is too small, previous task information cannot be fully stored, leading to stability loss (high FM). However, when $|Dpre|$ reaches around $1e5$, NBSP performs well and remains insensitive to buffer size beyond this point.

Table 20: Effect of the hyperparameter $|D_{pre}|$ on performance.

| $\mathbf{D}_{pre}$ | ASR ↑ | FM ↓ | FWT ↑ |
|---|---|---|---|
| 1e2 | $0.62 \pm 0.01$ | $0.99 \pm 0.01$ | $0.99 \pm 0.01$ |
| 1e3 | $0.62 \pm 0.01$ | $0.99 \pm 0.01$ | $0.98 \pm 0.01$ |
| 1e4 | $0.74 \pm 0.09$ | $0.67 \pm 0.21$ | $0.98 \pm 0.01$ |
| 1e5 | $0.95 \pm 0.05$ | $0.08 \pm 0.12$ | $0.98 \pm 0.01$ |
| 1e6 | $0.93 \pm 0.04$ | $0.13 \pm 0.13$ | $0.99 \pm 0.01$ |

$k$: We test values of $k$ ranging from 2 to 100 (see Table 21 below). When $k$ is small, frequent replay of previous experiences enhances stability but reduces plasticity (low FWT). In contrast, past experiences are underutilized, weakening stability. When $k$ is within the range of 5-13, NBSP performs well, demonstrating insensitivity to variations in $k$ in this range.

Table 21: Effect of the hyperparameter $k$ on performance.

| k | ASR ↑ | FM ↓ | FWT ↑ |
|---|---|---|---|
| 2 | $0.62 \pm 0.01$ | $0.02 \pm 0.02$ | $0.50 \pm 0.00$ |
| 5 | $0.95 \pm 0.04$ | $0.07 \pm 0.09$ | $0.97 \pm 0.02$ |
| 10 | $0.95 \pm 0.05$ | $0.08 \pm 0.12$ | $0.98 \pm 0.01$ |
| 13 | $0.94 \pm 0.04$ | $0.11 \pm 0.09$ | $0.98 \pm 0.01$ |
| 20 | $0.89 \pm 0.06$ | $0.21 \pm 0.13$ | $0.98 \pm 0.01$ |
| 100 | $0.66 \pm 0.01$ | $0.90 \pm 0.05$ | $0.99 \pm 0.01$ |

$\mathbf{T_{avg}}$: As shown in Table 22, when the window size is too small, the running averages of neuronal activations and performance become noisy, leading to inaccurate estimation of $\bar{a}$ and $\bar{q}$, which in turn hinders the correct identification of RL skill neurons. When the window size exceeds 50,000, NBSP becomes insensitive to the specific value, and performance remains consistently high. Since excessively large window sizes introduce unnecessary computational overhead, we choose 50,000 as a practical trade-off between estimation stability and efficiency.

Table 22: Effect of the hyperparameter $T_{avg}$ on performance.

| $\mathbf{T_{avg}}$ | ASR ↑ | FM ↓ | FWT ↑ |
|---|---|---|---|
| 5000 | $0.81 \pm 0.03$ | $0.44 \pm 0.08$ | $0.98 \pm 0.02$ |
| 25000 | $0.88 \pm 0.03$ | $0.30 \pm 0.08$ | $0.99 \pm 0.01$ |
| 50000 | $0.95 \pm 0.05$ | $0.08 \pm 0.12$ | $0.98 \pm 0.01$ |
| 100000 | $0.94 \pm 0.04$ | $0.09 \pm 0.11$ | $0.98 \pm 0.01$ |
| 150000 | $0.95 \pm 0.04$ | $0.08 \pm 0.11$ | $0.98 \pm 0.01$ |

In summary, there is a practical and systematic tuning criterion as follows:

- Stable performance within broad ranges.

  Each hyper-parameter exhibits a wide "safe range" within which NBSP performs consistently well. Therefore, tuning does not require fine-grained search. The recommended ranges are provided in Table 23.

Table 23: The recommended ranges of hyper-parameters.

| Hyper-parameters | Range |
|---|---|
| $m$ | $[0.15, 0.3]$ |
| $\alpha$ | $[0.1, 0.3]$ |
| $\|D_{\text{pre}}\|$ | $[10^5, \infty)$ |
| $k$ | $[5, 13]$ |

- Key hyper-parameter for task variation. Among the four hyper-parameters, $m$ is the primary one that may vary across tasks, as the number of RL skill neurons naturally depends on the skill complexity of each environment. In practice, tuning $m$ within $[0.15, 0.3]$ is sufficient in our experiments.

- Other hyper-parameters remain stable across benchmarks.

  The remaining three hyper-parameters require only minor adjustments within their recommended ranges. In our experiments, they remain fixed across all benchmarks, and NBSP still achieves strong and stable performance, demonstrating their robustness and confirming that the method does not rely on brittle manual tuning.

### C.10.7 TASK ORDER

In our experimental setup, we repeat each task twice within a cycling sequence to mitigate the potential influence of task order. For example, in a two-task cycling sequence (button-press-topdown → window-open), repeating these tasks results in the sequence (button-press-topdown → window-open → button-press-topdown → window-open). This sequence contains two occurrences of the subsequence (button-press-topdown → window-open) and one occurrence of (window-open → button-press-topdown), which helps mitigate the impact of task order. However, we acknowledge that this approach does not fully eliminate the influence of task order, and we will revise the description to clarify this point.

To further investigate the effect of task order, we conduct experiments with randomized task order. The results, shown in Table 24, indicate that task order does affect performance, particularly in terms of stability. However, the impact is modest, and NBSP still performs well in balancing stability and plasticity regardless of the task order. Determining task order presents a promising future direction, where task difficulty, diversity, and coherency might be taken into account.

Table 24: Performance of different task orders.

| Cycling sequential tasks | ASR ↑ | FM ↓ | FWT ↑ |
|---|---|---|---|
| (window-open → button-press-topdown) | $0.90 \pm 0.08$ | $0.17 \pm 0.13$ | $0.95 \pm 0.02$ |
| (button-press-topdown → window-open) | $0.95 \pm 0.05$ | $0.08 \pm 0.12$ | $0.98 \pm 0.01$ |
| (drawer-open → drawer-close) | $0.96 \pm 0.02$ | $0.07 \pm 0.06$ | $0.98 \pm 0.01$ |
| (drawer-close → drawer-open) | $0.92 \pm 0.05$ | $0.12 \pm 0.12$ | $0.97 \pm 0.01$ |

### C.10.8 PPO ALGORITHM

We further apply NBSP to PPO (Schulman et al., 2017) on a cycling task sequence (button-press-topdown→window-open). The results in Table 25 show that vanilla PPO performs worse than vanilla SAC in our setting, suffering from both stability and plasticity loss. NBSP helps reduce FM, improving stability and achieving a better balance, as reflected by a higher ASR. However, the effect is less pronounced than that of SAC. Potential reasons include:

Table 25: Comparison of vanilla PPO and PPO with NBSP.

| Method | ASR ↑ | FM ↓ | FWT ↑ |
|---|---|---|---|
| vanilla PPO | $0.40 \pm 0.04$ | $0.82 \pm 0.18$ | $0.66 \pm 0.12$ |
| PPO with NBSP | $0.49 \pm 0.06$ | $0.58 \pm 0.09$ | $0.67 \pm 0.11$ |

- On-Policy Nature of PPO: PPO is an on-policy algorithm, and cannot fully leverage the experience replay mechanism. While old experiences can still be sampled, they are more likely located outside the "trust region", leading to suboptimal updates.

- Differences in Exploration Mechanisms: SAC incorporates an entropy regularization term in its objective function. When NBSP masks RL skill neurons, the entropy term of SAC helps maintain exploration in other neurons, without sacrificing too much plasticity. In contrast, PPO's exploration is driven primarily by its stochastic policy and lacks explicit entropy constraint, making it more prone to instability if RL skill neurons are masked.

## D   ALGORITHM

The pseudo-code of the goal-oriented method to find RL skill neurons is presented in Algorithm 1. And the pseudo-code for SAC with NBSP is presented in Algorithm 2. Key differences from standard SAC are highlighted in blue. In addition to the extra input, two main modifications include the sampling process and the network update process.

---

**Algorithm 1** Procedure for Identifying RL Skill Neurons
---
**Input**: Initial average step $T_{avg}$, initial evaluation step $T$, initial proportion of RL skill neuron $m$, initial average activation $\overline{a}(\mathcal{N}) = 0$, initial average GM $\overline{q} = 0$, initial over-activation rate $R_{over} = 0$.

1: **for** each step $t$ **do**
2:     Compute activation $a(\mathcal{N}, t) \leftarrow \phi(\cdot)$
3:     Compute GM $q(t)$
4:     Compute average activation:

$$\overline{a}(\mathcal{N}) = \overline{a}(\mathcal{N}) + \frac{1}{T_{avg}} a(\mathcal{N}, t).$$

5:     Compute average GM:

$$\overline{q} = \overline{q} + \frac{1}{T_{avg}} q(t).$$

6: **end for**
7: **for** each step $t$ **do**
8:     Compute activation $a(\mathcal{N}, t) \leftarrow \phi(\cdot)$
9:     Compute GM $q(t)$
10:     Capture association:

$$R_{over} = R_{over} + \frac{1}{T} \mathbb{1}_{[\mathbb{1}_{[a(\mathcal{N},t) > \overline{a}(\mathcal{N})]} = \mathbb{1}_{[q(t) > \overline{q}]}]}$$

11: **end for**
12: Derive scores $Score$ for each neuron:

$$Score(\mathcal{N}) = max(R_{over}(\mathcal{N}), 1 - R_{over}(\mathcal{N}))$$

13: Identify the top-performing neurons as RL skill neurons:

$$\mathcal{N}_{RL\ skill} = \tau_m(Score(\mathcal{N}))$$

---

---

**Algorithm 2** Neuron-level Balance between Stability and Plasticity (NBSP) Applied in SAC

---

Initialize policy parameters $\theta$, Q-function parameters $\phi_1$, $\phi_2$, and target Q-function parameters $\phi_1'$, $\phi_2'$

Initialize empty replay buffer $\mathcal{D}$

Initialize replay interval $k$

**Input: Replay buffer $\mathcal{D}_{\mathbf{pre}}$, mask of the policy $\mathbf{mask}_\theta$ and mask of the Q-function parameters $\mathbf{mask}_{\phi_1}, \mathbf{mask}_{\phi_2}$**

  1: **for** each task **do**

  2:     **for** each iteration **do**

  3:         **for** each environment step **do**

  4:             Sample action $a_t \sim \pi_\theta(a_t|s_t)$

  5:             Execute action $a_t$ and observe reward $r_t$ and next state $s_{t+1}$

  6:             Store $(s_t, a_t, r_t, s_{t+1})$ in replay buffer $\mathcal{D}$

  7:         **end for**

  8:         **for** each gradient step **do**

  9:             **if step $\equiv 0 \pmod{\mathbf{k}}$ then Sample batch of transitions $(\mathbf{s_i, a_i, r_i, s_{i+1}})$ from $\mathcal{D}_{\mathbf{pre}}$**

10:             **else** Sample batch of transitions $(s_i, a_i, r_i, s_{i+1})$ from $\mathcal{D}$

11:             **end if**

12:             Compute target value:

$$y_i = r_i + \gamma\left(\min_{j=1,2} Q_{\phi_j'}(s_{i+1}, \tilde{a}_{i+1}) - \alpha \log \pi_\theta(\tilde{a}_{i+1}|s_{i+1})\right), where\ \tilde{a}_{i+1} \sim \pi_\theta(\cdot|s_{i+1})$$

13:             Update Q-functions by one step of gradient descent with mask:

$$\phi_j \leftarrow \phi_j - \lambda_Q \mathbf{mask}_{\phi_j} \nabla_{\phi_j} \frac{1}{N} \sum_i \left(Q_{\phi_j}(s_i, a_i) - y_i\right)^2 \quad \text{for } j = 1, 2$$

14:             Update policy by one step of gradient ascent with mask:

$$\theta \leftarrow \theta + \lambda_\pi \mathbf{mask}_\theta \nabla_\theta \frac{1}{N} \sum_i \left(\alpha \log \pi_\theta(a_i|s_i) - \min_{j=1,2} Q_{\phi_j}(s_i, a_i)\right)$$

15:             Update temperature $\alpha$ by one step of gradient descent:

$$\alpha \leftarrow \alpha - \lambda_\alpha \nabla_\alpha \frac{1}{N} \sum_i \left(-\alpha \log \pi_\theta(a_i|s_i) - \alpha\bar{\mathcal{H}}\right)$$

16:             Update target Q-function parameters:

$$\phi_j' \leftarrow \tau\phi_j + (1 - \tau)\phi_j' \quad \text{for } j = 1, 2$$

17:         **end for**

18:     **end for**

19:     **Select RL skill neurons $\{\mathcal{N}_{\mathbf{RL\ skill}}\}$ according to Algorithm 1**

20:     **Update $\mathbf{mask}_{\phi_1}, \mathbf{mask}_{\phi_2}$ and $\mathbf{mask}_\theta$:**

$$mask(\mathcal{N}) = \begin{cases} \alpha(1 - Score(\mathcal{N})) & \textbf{if } \mathcal{N} \in \mathcal{N}_{RL\ skill} \\ 1 & \textbf{if } \mathcal{N} \notin \mathcal{N}_{RL\ skill} \end{cases}$$

21:     **Store part of $\mathcal{D}$ into $\mathcal{D}_{\mathbf{pre}}$**

22: **end for**

---

# E   Limitation and Future Work

**Limitation**. While the proposed NBSP method effectively balances stability and plasticity in DRL, it does have a notable limitation. Specifically, the number of RL skill neurons must be manually determined and adjusted according to the complexity of the learning task, as there is no automatic mechanism for this selection. And our method currently faces challenges when applied to longer task sequences (e.g., 10+ tasks). One key limitation is the constraint imposed by the model scale,

which inherently limits the number of skills it can learn. As the number of tasks increases, the overlap between skill neurons across different tasks may become significant. Consequently, applying a mask to protect RL skill neurons can restrict the learning of new tasks, making it difficult to scale without introducing interference with previously learned knowledge.

**Future work**. The neuron analysis introduced in this work offers a novel approach for identifying RL skill neurons, significantly enhancing the balance between stability and plasticity in DRL. The identification of RL skill neurons opens up several promising directions for future research and applications, such as: (1) Model Distillation: by focusing on RL skill neurons, it becomes possible to distill models by pruning less relevant neurons, leading to more efficient and compact models with minimal performance degradation. (2) Bias Control and Model Manipulation: RL skill neurons could be leveraged to control biases and modify model behaviors by selectively adjusting their activations. This approach could be particularly valuable in scenarios requiring specific outputs or behaviors.

While our current method may not yet fully address longer task sequences, it lays a strong foundation for future research. Moving forward, we aim to explore strategies to better leverage RL skill neurons for continual learning over an extended sequence of tasks. What's more, its applicable potential extends beyond DRL. It could also be adapted to other learning paradigms, such as supervised and unsupervised learning, to address similar stability-plasticity challenges. In future work, we plan to explore these extensions and verify their effectiveness across various domains.

# F THE USE OF LARGE LANGUAGE MODELS

In preparing this manuscript, we employ the large language model solely for minor linguistic refinement. The LLM is not used for research design, data analysis, result interpretation, or generating any scientific content. All conceptual contributions, experimental designs, analyses, and conclusions are entirely the work of the authors.

