# OpenReview forum: "NBSP: A Neuron-Level Framework for Balancing Stability and Plasticity in Deep Reinforcement Learning"
_ICLR.cc/2026/Conference — Submitted to ICLR 2026_

### Official Review · Reviewer_t1NL · 2025-10-29

**Soundness:** 2
**Presentation:** 4
**Contribution:** 3
**Rating:** 4
**Confidence:** 4

**Summary:**

This paper proposes a method for helping with the stability-plasticity tradeoff on a neuron-level basis. It proposes a metric to identify the neurons that are crucial for performance, and uses gradient masking and replay on top to preserve the already learned patterns.
The paper performs experiments on Metaworld, Atari, and DMC benchmarks, and compares their method with multiple methods proposed in continual learning, across multiple metrics, and shows that their approach (NBSP) can outperform without the addition of parameters or using complex networks.

**Strengths:**

1. The paper is well-written, and there is a natural flow for related work and methodology.

2. The paper is well-motivated, and the proposed methodology does in fact avoid complex NN designs and the use of large networks, which are crucial benefits in deep RL.

3. The paper leverages the use of multiple evaluation metrics for plasticity, stability, and overall performance for assessing their method and comparing it to others. Ablation studies explore the components of the proposed method thoroughly, and also the use of the algorithm in other agents than SAC is included.

**Weaknesses:**

1. The score does not capture what the paper intends to.  $1 - R_{\text{over}}(\mathcal{N})$ does not only include neurons whose activity hold a negative correlation with performance, as stated and intended in line 233,  but also other cases:
$$
1 - R_{\text{over}}(N)
= 1 - \frac{\sum_{t=1}^{T} 1\Big[ 1[a(N, t) > \bar{a}(N)] = 1[q(t) > \bar{q}] \Big]}{T}
$$

$$
= \frac{T - \sum_{t=1}^{T} 1\Big[ 1[a(N, t) > \bar{a}(N)] = 1[q(t) > \bar{q}] \Big]}{T}
$$

$$
= \frac{\sum_{t=1}^{T} \Big( 1 - 1\big[ 1[a(N, t) > \bar{a}(N)] = 1[q(t) > \bar{q}] \big] \Big)}{T}
$$

where the numerator is:
$$
\begin{cases}
0, & \text{if } a(N,t) > \bar{a}(N) \text{ and } q(t) > \bar{q},\\
1, & \text{otherwise.}
\end{cases}
$$
and "otherwise" includes all the other cases, i.e., (a < ā, q > q̄), (a > ā, q < q̄), (a < ā, q < q̄)
whereas, as stated in the paper, it was intended to account for only  (a < ā, q > q̄), meaning " when the activation of a neuron falls below its average activation, the agent performs well."


2. Three seeds are not enough to make strong claims about the performance of the method compared to the others. Can you take a few of the methods in one of the settings and run more seeds for them to strengthen the claims?

3. Although the paper does a great job of exploring the effect of the NSBP hyperparameters on the model performance, other methods’ hyperparameters are underexplored. How are the other baselines' hyperparameters tuned? Are they using their default hyperparameters? If so, it might not be fair to use those hypers as they were found for the settings they were experimented on. In particular, is COTASP tuned, since it consistently underperforms in all the metaworld benchmarks? What is the reason for its performance collapse?

4. The paper emphasizes the distinction between CRL and DRL. However, the experiments are mostly done in settings designed for CRL ( non-stationary or task change in the environment). Can you elaborate on the reason for this emphasis and its relation to your method?

**Questions:**

1. The average goal metric works for settings where there is a specific goal or an episode. How does your method and this metric extend to continuing settings where there are no episodes, or a clear/binary notion of success?

2. How do the experiment results change after fixing the score function?

---

> ### Author Response · Authors · 2025-11-19
>
> We appreciate your valuable feedback on the score function, experiment seed, hyperparameters of other baselines and your acknowledgement of novelty, writing, and experimental validation of our paper. And we will address each of your comments and concerns in the following responses.
>
> Q1: The score does not capture what the paper intends to
>
> A1: Thank you for your suggestion for better clarification. 0ur intention is to distinguish between positive and negative relationships between neuron activation and agent performance. To better clarify:
>
> - The condition $(1_{[a(\mathcal{N}, t) > \overline{a}(\mathcal{N})]} == 1_{[q(t) > \overline{q}]}) = 1$ covers the cases $(a > \overline{a}, q > \overline{q})$ and $(a < \overline{a}, q < \overline{q})$, where activation and performance change in the **same direction** (positive correlation).
> - The condition $(1_{[a(\mathcal{N}, t) > \overline{a}(\mathcal{N})]} == 1_{[q(t) > \overline{q}]}) = 0$ covers the cases $(a < \overline{a}, q > \overline{q})$ and $(a > \overline{a}, q < \overline{q})$, where activation and performance change in **opposite directions** (negative correlation).
>
> And “when the activation of a neuron falls below its average activation, the agent performs well” refers specifically to the negative-correlation cases $(a < \overline{a}, q > \overline{q})$ and $(a > \overline{a}, q < \overline{q})$. We will revise the text to clarify this point.
>
> Q2: seeds
>
> A2: Thank you for your suggestion. To improve the reliability of our results, **we extend the number of random seeds from 3 to 5** for the methods(NBSP, vanilla SAC, and NE) on the task sequence (window-open  $\rightarrow$ window-close $\rightarrow$ drawer-open $\rightarrow$ drawer-close) with the highest performance variance in Meta-World. The updated results over five seeds are reported in the table below. While using more seeds improves statistical reliability, **the overall trends remain consistent and do not change our main conclusions.**
>
> |             | ASR $\uparrow$ | FM $\downarrow$ | FWT $\uparrow$ |
> | ----------- | -------------- | --------------- | -------------- |
> | Vanilla SAC | 0.31 ± 0.08    | 0.81 ± 0.14     | 0.57 ± 0.27    |
> | NE          | 0.55 ± 0.04    | 0.68 ± 0.08     | 0.89 ± 0.05    |
> | NBSP        | 0.66 ± 0.10    | 0.49 ± 0.14     | 0.89 ± 0.08    |
>
> Q3: hyper-parameters of other baselines
>
> A3: Thank you for the question. For all baseline methods, we adopt their **default hyperparameter settings** as reported in their original papers or official implementations. Due to computational resource constraints, it is infeasible for us to exhaustively tune the full hyperparameter space for each baseline, and we acknowledge that the performance of the baselines can be better if we further tune their hyper-parameters.
>
> Specifically, regarding COTASP, we conduct additional tuning on its sparsity hyperparameter within the search range specified in the original paper. Even with tuning, COTASP still performs poorly in our setting.
> We suspect its performance degradation may stem from two main factors:
>
> 1. **Task similarity and simple task descriptions in Meta-World.**
>    The task embeddings lack clear separability, making it difficult for the dictionary to learn meaningful semantic relationships across tasks.
> 2. **Mismatch between task characteristics and the learned dictionary.**
>    The distribution of skills required in our tasks differs from the knowledge captured by COTASP’s pre-learned dictionary because the tasks in our experiments are different from the tasks in COTASP, which prevents the method from generating effective sparse prompts.

---

> ### Author Response · Authors · 2025-11-19
>
> Q4: the distinction between CRL and DRL
>
> A4: Thank you for the question. Our emphasis on the distinction between CRL and DRL is intentional and closely related to the design of our method.
> Most CRL works focus on long task sequences (often 10+ tasks), and the core challenge is the stability–plasticity dilemma, which is also the main focus of our research. **As an initial exploration, we use short cyclic sequences in DRL environments to systematically examine whether RL Skill Neurons can balance stability and plasticity.** Such non-stationary settings act as a minimal testbed for the stability–plasticity dilemma, making them suitable for evaluating our mechanism without the confounding complexity of full CRL. The results show that RL Skill Neurons exhibit strong potential for balancing stability and plasticity, which is required in CRL. **Extending NBSP and RL Skill Neurons to full CRL benchmarks with long task sequences is a natural next step of our research.**
>
> Q5: How does your method and this metric extend to continuing settings
>
> A5: Thank you for the question. You are correct that the average Goal Metric (GM) in our paper is defined for episodic RL settings with a clear notion of success (e.g., success rate or episode return).
>
> For continuing RL settings, although episodes or binary success signals may be absent, NBSP can still be naturally extended by redefining the GM in a way that reflects task performance under the continuing setting:
>
> 1. **Discounted return as GM:**
>    In the absence of episodic boundaries, the **discounted return** over a fixed horizon or sliding window can serve as a continuous performance indicator.
> 2. **Constraint satisfaction rate:**
>    For tasks defined by ongoing constraints or control objectives (e.g., maintaining balance, keeping energy consumption low), the GM can be computed as the proportion of time steps that satisfy the task-specific constraints. This generalizes the “success rate” to non-episodic environments.
>
> These alternatives maintain the core requirement of NBSP: a scalar performance metric that correlates with the agent’s behavioral quality. Therefore, while the GM in our experiments is defined for episodic settings, the method itself is not limited to them and can be adapted to continuing RL scenarios by substituting an appropriate performance signal.

---

> ### Comment · Reviewer_t1NL · 2025-11-27
>
> Thank you for your comments and clarifications.  The clarification of the score function makes it now clearer.
>
> I would like to follow up regarding the interpretation of the score and the definition of skill neurons.
>
> For instance, for the positive-correlation case, it is not empirically obvious that a neuron that becomes more active when performance is high (i.e., satisfies ( a > ā, q > q̄)) will also become less active when performance is low (i.e., ( a < ā, q < q̄) ). In other words, the current formulation seems to implicitly assume that neurons satisfying the first condition also satisfy the second, but this may not necessarily hold. A similar concern applies to the negative-correlation case.
>
> Because the setting is continual, activation–performance relationships may shift over time, which makes this assumption more critical.
>
> Could you clarify, or empirically show that neurons that satisfy ( a > ā, q > q̄) are the same neurons that hold (a < ā, q < q̄) during an experiment?

---

> > ### Author Response · Authors · 2025-11-29
> >
> > Thank you for your insightful question regarding the interpretation of the score and the definition of RL Skill neurons. To further clarify this point, we conduct an additional analysis to examine whether neurons that satisfy the condition $(a > \bar{a}, q > \bar{q})$ also tend to satisfy $(a < \bar{a}, q < \bar{q})$ during the neuron-identification process. Specifically, for each neuron, we record the proportion of occurrences in the four possible activation–performance cases, denoted as $p_{>>}, p_{<<}, p_{><}, p_{<>}$. We then focus on neurons whose combined proportion $p_{>>} + p_{<<}$ exceeds 0.5, and compute their average proportion $\mathbb{E}(p_{>>} + p_{<<})$. An analogous analysis is also conducted for the negative-correlation cases $p_{><} + p_{<>}$.
> >
> > We report the summary statistics separately for RL skill neurons and for all neurons, as shown in the table below. The results indicate that for the identified RL skill neurons, the average proportion of $(a > \bar{a}, q > \bar{q})$ or $(a < \bar{a}, q < \bar{q})$ is very high (0.91) when the combined proportion exceeds 0.5. This suggests that neurons that become more active when performance is high also tend to become less active when performance is low as the average proportion of the opposite-pattern cases $(a > \bar{a}, q < \bar{q})$ or $(a < \bar{a}, q > \bar{q})$ is only 0.09. In contrast, this property does not consistently hold when considering all neurons, which show a much higher average proportion (0.32) for the opposite-pattern cases. This suggests that the assumption only targets RL Skill neurons and does not generally hold across the whole neurons. A similar trend is observed for the negative-correlation cases as well.
> >
> > These findings confirm that the activations of RL Skill neurons play an important role in determining the network’s performance, demonstrating that the identified neurons consistently encode critical knowledge.
> > |                  | $\mathbb{E}(p_{>>} + p_{<<})$ | $\mathbb{E}(p_{><} + p_{<>})$ |
> > | ---------------- | ----------------- | ----------------- |
> > | RL Skill neurons | 0.91              | 0.89              |
> > | The whole neurons      | 0.68              | 0.64              |

---

### Official Review · Reviewer_4tP5 · 2025-10-30

**Soundness:** 2
**Presentation:** 3
**Contribution:** 2
**Rating:** 4
**Confidence:** 3

**Summary:**

This paper introduces a neural-level method to overcome catastrophic forgetting issues in deep reinforcement learning (DRL). They propose to both mask the gradient of task-specific neurons and train the neural network with multi-task experience replay. Experiments show improved results compared to baselines on a DRL benchmark.

The paper is well-written, the method is novel and the preliminary results are promising. My main concern is about the experimental protocol. First, the authors compare their approach to baselines of the field only in the Meta-World benchmarks. Second, there is no clear reasons for not testing the method on supervised benchmarks that are more widely used in continual learning, such as adapted versions of cifar100 and imagenet. I suspect that the "Experience replay only" version will achieve much better performance in supervised learning. In addition, the authors do not provide a clear analysis of why Experience replay and neuron masking are complementary; the bad results of "Experience replay only" require explanations. Third, it is unclear how the method scales when considering more than four tasks.

The authors should address these issues before resubmission.

Minor comments:
- "reasonably broad range of values (from 0.15 to 0.3)", it is unclear why the authors mention this specific range, given the shape of the curve in Figure 4.
- It is unclear why increasing the number of masked neurons "compromises their learning capacity and causes the true RL skill neurons to adjust their activations to accommodate new tasks, ultimately reducing stability".

**Strengths:**

The paper is well-written, the approach is novel and relevant

**Weaknesses:**

Experiments are insufficient

**Questions:**

Please, see above.

---

> ### Author Response · Authors · 2025-11-19
>
> We appreciate your valuable feedback on the experimental protocol, description and your acknowledgement of writing, novelty, experiment results of our paper. We will carefully address each of your comments below.
>
> Q1: only in the Meta-World benchmark
>
> A1: Thank you for pointing this out. Our comparisons are not limited to Meta-World. **In addition to the Meta-World experiments, we also evaluate NBSP against the same set of baselines on the Atari and DMC benchmarks**, and the corresponding results are reported in Table 5 of the paper.
> Meta-World is one of the most common and challenging benchmarks for multi-task RL, featuring sparse rewards, diverse manipulation skills, and multi-task sequences used in prior works. Atari, while simpler in visual dynamics, provides a discrete action space, allowing us to validate that NBSP generalizes beyond continuous-control settings. DMC is built on the MuJoCo physics engine and introduces complex continuous control, rich dynamics, and perceptual disturbances. These characteristics make DMC a widely recognized and challenging benchmark in deep RL studies.
> Across these benchmarks, NBSP consistently demonstrates strong performance and generalization, confirming that its advantages are not specific to a single benchmark family.
>
> Q2: supervised benchmarks
>
> A2: Thank you for the thoughtful question. We would like to clarify that our work is specifically designed to address the **stability–plasticity dilemma in reinforcement learning (RL)**. Although RL and supervised continual learning share conceptual similarities, their learning paradigms differ substantially. Our method is evaluated in Meta-World, Atari, and DMC benchmarks, which are widely used to study stability, plasticity, and catastrophic forgetting in the RL setting [1,2,3].
> In contrast, CIFAR100 and ImageNet are **supervised learning datasets** and cannot be directly used as RL environments without extensive reformulation.
>
> **Conceptually, NBSP can be extended to supervised continual learning. However, the skill neuron identification module would need to be adapted**, as the current version relies on a Goal Metric (GM) defined in terms of success rate or average return in online RL. Exploring these adaptations is an interesting future direction.
>
> [1] Ahn H, Hyeon J, Oh Y, et al. Reset & distill: A recipe for overcoming negative transfer in continual reinforcement learning[J]. arXiv preprint arXiv:2403.05066, 2024.
>
> [2] He J, Li K, Zang Y, et al. Not all tasks are equally difficult: Multi-task deep reinforcement learning with dynamic depth routing[C]//Proceedings of the AAAI Conference on Artificial Intelligence. 2024, 38(11): 12376-12384.
>
> [3] Lewandowski A, Bortkiewicz M, Kumar S, et al. Learning Continually by Spectral Regularization[C]//The Thirteenth International Conference on Learning Representations. OpenReview. net, 2025.
>
> Q3: why Experience replay and neuron masking are complementary
>
> A3: Thank you for raising this important point. Experience replay and gradient masking address different types of interference, and their complementary roles are crucial for achieving stability–plasticity balance.
>
> 1. **Gradient masking prevents destructive updates to task-critical neurons.**
>  Gradient masking regulates **how** the parameters are updated by selectively blocking updates to RL skill neurons.
>  This protects task-specific representations from being overwritten when learning new tasks, directly reducing gradient interference.
> 2. **Experience replay regulates the distribution of training data.**
>  Experience replay determines **what** data the agent learns from, balancing samples from the current task and previous tasks.
>  By reintroducing transitions from earlier tasks, experience replay keeps the overall feature space within the range required to maintain previously learned behaviors.
> 3. Why they are complementary.
>    - When replayed samples from old tasks arrive, **gradient masking ensures that primarily non–RL skill neurons are updated**, preventing the model from drifting toward the new task distribution.
>    - Conversely, **gradient masking alone would cause old-task representations to stagnate or drift if only new-task data are seen, experience replay provides the necessary signal from past tasks**.
> 4. Why “Experience Replay only” does not fully resolve forgetting.
>  Our results in Table 2 show that Experience replay alone does not perform poorly. In fact, It improves stability compared to vanilla SAC (lower FM from 0.99 to 0.50) and yields a better stability–plasticity trade-off (higher ASR from 0.62 to 0.70). However, experience replay alone cannot prevent destructive updates to the RL skill neurons during new-task learning. As these neurons are repeatedly used for representing high-level task structure, even small updates accumulate and degrade past-task performance. **Experience replay can replay old data, but it cannot undo harmful parameter shifts once they occur.**

---

> ### Author Response · Authors · 2025-11-19
>
> Q4: longer task sequences
>
> A4: Thank you for your question. To further evaluate NBSP under longer task sequences, we additionally conduct experiments on a 10-task sequence (button-press-topdown $\rightarrow$ window-close $\rightarrow$ door-open $\rightarrow$ drawer-close $\rightarrow$ drawer-open $\rightarrow$ door-close $\rightarrow$ button-press-topdown-wall $\rightarrow$ window-open $\rightarrow$ push $\rightarrow$ reach) in the Meta-World benchmark. As shown in the table below, NBSP continues to outperform vanilla SAC by a large margin, particularly in terms of stability, where the forgetting metric decreases from 0.79 to 0.32.
>
> We also compare NBSP with NE, which is the strongest baseline in our earlier Meta-World experiments. Although NE achieves a better stability–plasticity balance than vanilla SAC, it still lags behind NBSP on the longer 10-task sequence.
> These results demonstrate that **NBSP generalizes well beyond short task chains and is effective even in long-horizon task settings.**
>
> |             | ASR $\uparrow$ | FM $\downarrow$ | FWT $\uparrow$ |
> | ----------- | -------------- | --------------- | -------------- |
> | vanilla SAC | 0.27 ± 0.05    | 0.79 ± 0.07     | 0.52 ± 0.19    |
> | NE          | 0.58 ± 0.05    | 0.44 ± 0.04     | 0.64 ± 0.03    |
> | NBSP        | 0.66 ± 0.02    | 0.32 ± 0.06     | 0.74 ± 0.01    |
>
> Q5: reasonably broad range of values (from 0.15 to 0.3)
>
> A5: Thank you for the question. We refer to the range 0.15–0.3 because ASR remains consistently above 0.9 within this interval, as shown in Figure 4, indicating that NBSP is highly robust to the choice of $m$ across this region. **Our intention is not to claim that this is the only valid range, but rather to highlight a stable interval in which performance is reliably strong and insensitive to small changes in $m$.** This illustrates that NBSP does not require precise tuning of $m$; instead, selecting any value within this interval yields similarly high performance, demonstrating the robustness of the method.
>
> Q6: increasing the number of masked neurons
>
> A6: Thank you for pointing this out. We have revised it as follows:
>
> > Specifically, when the proportion of masked neurons is too small, not all neurons that encode task-specific skills can be correctly identified. As a result, the true RL skill neurons must adjust their activations to accommodate new tasks, ultimately reducing stability. Conversely, when the proportion becomes too large, neurons that do not encode task-specific skills may be mistakenly selected as RL skill neurons, which compromises their ability to learn new tasks and thereby reduces plasticity

---

> > ### Comment · Reviewer_4tP5 · 2025-11-20
> > **Response**
> >
> > I thank the authors for their answers and clarifications.
> >
> > A2: " In contrast, CIFAR100 and ImageNet are supervised learning datasets and cannot be directly used as RL environments without extensive reformulation. [...] the skill neuron identification module would need to be adapted, as the current version relies on a Goal Metric (GM) defined in terms of success rate or average return in online RL. [...]. "
> >
> > One could replace the success rate in achieving a task by the success rate in categorizing an augmented image. It does not seem that extensive.
> >
> > A3: I find the answer of the authors very speculative, can they back up their statements with experimental or theoretical evidence ? This could be done by quantifying the drift of neurons that would have been selected by the masking mechanism (but are not), and correlating this drift with the decrease of performance. This could also be compared with the drift that potentially occurs in supervised training (CIFAR100).
> >
> > Characterizing the weakness of experience replay alone, even (or especially) if specific to DRL, could considerably strengthen the paper.
> >
> > This is especially important as most comparison baselines do not use experience replay and could easily be enhanced with experience replay. Could the authors comment on the fairness of the comparison ?
> >
> > A5: My mistake stems from the ticks of the x-axis, I suggest that the authors increase the number of ticks.

---

> ### Author Response · Authors · 2025-11-24
>
> Q2: CIFAR100 and ImageNet
>
> A2: Thank you for the question. For supervised learning datasets such as CIFAR-100 or ImageNet, one could indeed define “success” as correct image classification when computing the Goal Metric (GM). This may provide an interesting way to identify skill neurons in supervised settings.
> However, our method is fundamentally developed for reinforcement learning, where the agent is trained through the interaction loop of state → action → reward, and NBSP is built on the SAC framework. In supervised learning, this interaction structure does not exist because there are no states, actions and reward signals. Therefore, NBSP cannot be directly applied to these datasets without redefining the underlying training paradigm.
> Exploring how to construct an analogous framework for supervised learning, and whether skill neuron identification generalizes beyond RL, is an interesting future research direction.
>
> Q3: experimental  evidence
>
> A3: Thank you for the insightful suggestion. We agree that validating the mechanism beyond qualitative intuition is important.
>
> i. Following your advice, we conduct an additional experiment to quantify how sensitive the network performance is to perturbations on neurons that would be masked by NBSP (i.e., RL Skill neurons) versus those that would not (non–RL Skill neurons). Specifically, in the button-press-topdown task, we inject zero-mean Gaussian noise into the activations of 10% of neurons. The noise is added either to (1) RL Skill neurons or (2) non-RL Skill neurons, with the same noise magnitude and proportion in both settings. The results are reported in the table below. We observe the following:
>
> 1. Perturbing RL Skill neurons causes a drastic performance drop, showing that these neurons are indeed critical for representing task-specific knowledge.
> 2. Perturbing non-RL Skill neurons has only mild impact on the final performance.
> 3. This difference directly confirms that masking high-score neurons (RL Skill neurons) is necessary for maintaining learned behavior, supporting the stability side of the stability–plasticity balance.
>
> |              | add noise to the the activation of RL Skill neurons | add noise to the the activation of non-RL Skill neurons |
> | ------------ | --------------------------------------------------- | ------------------------------------------------------- |
> | success rate | 0.22 ± 0.03                                         | 0.92 ± 0.02                                             |
>
> ii. To more clearly characterize the limitations of experience replay and its relation to our method, we conduct additional analyses focusing on gradient-norm variance and parameter drift.
>
> - Gradient-norm variance
>
>   We first summarize the gradient-norm variance under four settings: vanilla SAC, SAC combined with gradient masking, SAC combined with experience replay, and NBSP. The results are shown in the table below. We observe the following ordering of gradient-norm variance:
>   $$
>   \text{SAC + experience replay}  >  \text{SAC}  >  \text{NBSP}  > \text{SAC + gradient masking}.
>   $$
>   This ordering highlights a fundamental weakness of experience replay: although experience replay introduces past experiences that are beneficial for memory retention, it simultaneously injects high-variance gradients due to stale, off-policy, and often task-mismatched samples. These noisy gradients destabilize optimization and make updates more susceptible to drifting toward suboptimal directions, especially when switching to new tasks. As a result, experience replay alone can degrade the agent’s plasticity, precisely because the instability of these gradients interferes with learning new behaviors efficiently.
>
>   In contrast, gradient masking systematically reduces gradient-norm variance by suppressing interfering gradient components. When combined with experience replay, gradient masking effectively cancels out the instability caused by experience replay. The resulting variance is not only far lower than SAC + experience replay, but even below vanilla SAC. Thus, NBSP retains the benefits of experience replay while eliminating the main weakness.
>
>   |                        | vanilla SAC | NBSP | SAC + gradient masking | SAC + experience replay |
>   | :--------------------: | :---------: | :--: | :--------------------: | :---------------------: |
>   | gradient-norm variance |    7.45     | 5.62 |          3.38          |          8.96           |

---

> ### Author Response · Authors · 2025-11-24
>
> - Parameter-drift
>
>   We further evaluate the cosine similarity between the network parameters before and after learning a new task. The ordering of parameter similarity is:
>   $$
>   \text{NBSP} > \text{SAC + gradient masking} > \text{SAC + experience replay} > \text{vanilla SAC}.
>   $$
>   Both experience replay and gradient masking help constrain parameter drift, but in different ways:
>
>   - Experience replay provides historical information that anchors parameters, but stale samples still generate conflicting gradient updates that cause substantial displacement.
>   - Gradient masking regulates harmful directions in the gradient space, but without explicit historical supervision it cannot fully prevent forgetting.
>
>   As a result, neither experience replay nor gradient masking alone is sufficient. experience replay alone still allows large drifts, and gradient masking alone cannot preserve full historical knowledge. When combined, however, experience replay supplies the required past-task information while gradient masking stabilizes the update directions. This complementary effect ensures that parameter changes stay within a range that preserves previously learned behaviors. This explains why NBSP achieves significantly stronger stability–plasticity performance than either mechanism individually.
>
>   |                    | vanilla SAC | NBSP | SAC + gradient masking | SAC + experience replay |
>   | :----------------: | :---------: | :--: | :--------------------: | :---------------------: |
>   | network similarity |    0.56     | 0.89 |          0.79          |          0.75           |
>
>
>
> iii. Experience replay is indeed a key component of NBSP. Our method is designed such that experience replay and gradient masking operate synergistically. This cooperation is essential for achieving the strong stability–plasticity balance observed in our experiments.
> Regarding fairness of comparison, we follow the baselines as originally proposed. Many prior  works intentionally avoid experience replay due to their design principles or assumptions (e.g., concerns about non-stationary environments, or additional variance). Therefore, evaluating our method against the official baseline implementations is standard and fair practice.
>
> To further validate that NBSP’s advantage is not merely due to the inclusion of experience replay, we additionally incorporate experience replay into several baseline methods. The results are summarized in the table below. While experience replay generally improves stability for these baselines to some extent, their performance still does not reach that of NBSP. Importantly, adding experience replay does not change any of the main conclusions drawn in the paper. Thus, although experience replay can enhance some baselines, the substantial performance gap indicates that NBSP’s improvements are not attributable to experience replay alone, but rather to the coordinated interaction between experience replay and our adaptive gradient masking mechanism.
>
> |                        | ASR $\uparrow$ | FM $\downarrow$ | FWT $\uparrow$ |
> | ---------------------- | -------------- | --------------- | -------------- |
> | ANCL                   | 0.61±0.01      | 0.95±0.05       | 0.95±0.03      |
> | ANCL+experience replay | 0.83±0.07      | 0.42±0.18       | 0.97±0.02      |
> | CoTASP                 | 0.03±0.00      | 0.01±0.00       | 0.04±0.01      |
> | NE                     | 0.60±0.03      | 1.00±0.00       | 0.94±0.05      |
> | NE+experience replay   | 0.71±0.03      | 0.73±0.10       | 0.96±0.01      |
> | UPGD                   | 0.39±0.03      | 0.82±0.05       | 0.54±0.07      |
> | UPGD+experience replay | 0.51±0.06      | 0.68±0.14       | 0.71±0.13      |
> | NBSP                   | 0.95±0.05      | 0.08±0.12       | 0.98±0.01      |
>
> Q5: the number of ticks
>
> A5: Thank you for pointing this out. The misunderstanding indeed arose from the sparse x-axis tick marks. We have updated the figures in the resubmission by increasing the number of ticks to improve readability and avoid potential misinterpretation.

---

> > ### Comment · Reviewer_4tP5 · 2025-11-25
> > **Response to authors**
> >
> > I thank the authors for the additional experiments, which clarify the role and necessity of gradient masking. I also appreciate the provided experiments combining the baselines with experience replay. These results shed light on the limitations of Experience Replay and highlight the advantages of the proposed method. I will increase my score to 6 once this analysis is added to the paper.
> >
> > The reason I am not increasing the score to 8 is that I am not convinced by the argument that the proposed method cannot be easily applied to supervised learning, as the interaction structure of RL plays no role in the proposed contribution. To clarify, I am not asking for a strong supervised-learning comparison that demonstrates quantitative advantages. The variance of the gradients and the weight-drift behavior may indeed be specific to RL; if that is the case, one would expect NBSP to offer no significant advantages over using "experience replay alone". Regardless of the outcome, such an experiment would help determine whether the issues encountered by experience replay alone are specific to RL, and it would clarify the generality of the method.

---

> > > ### Author Response · Authors · 2025-11-26
> > >
> > > Thank you for your insightful suggestion. We have incorporated the analysis regarding the  gradient masking and experience replay into the latest revision of the paper. Concerning the generalization of NBSP to supervised learning, we agree that this is an meaningful direction. And we will explore this extension in future work to further clarify the generality and applicability of the proposed method.

---

### Official Review · Reviewer_QyWf · 2025-10-31

**Soundness:** 3
**Presentation:** 3
**Contribution:** 3
**Rating:** 4
**Confidence:** 3

**Summary:**

This paper introduces NBSP (Neuron-level Balance between Stability and Plasticity), a framework that tackles the stability–plasticity dilemma in deep reinforcement learning (DRL). Unlike prior methods that regulate stability and plasticity at the network or parameter level, NBSP operates at the neuron level. It identifies RL skill neurons, neurons whose activations correlate strongly with task goals (e.g., success rate or return), and protects them during new task learning. NBSP combines Goal-oriented neuron identification to detect neurons critical for knowledge retention; and Adaptive gradient masking and experience replay, selectively applied to those neurons.

Experiments on Meta-World, Atari, and DMC benchmarks show that NBSP improves the trade-off between stability and plasticity, achieving higher Average Success Rate (ASR), lower Forgetting Measure (FM), and higher Forward Transfer (FWT) than nine competitive baselines (for example EWC, NPC, CoTASP, NE, UPGD). Ablation studies confirm the complementary roles of gradient masking and experience replay, and demonstrate that the goal-oriented neuron identification strategy is crucial to performance.

**Strengths:**

- The neuron-level approach to balancing stability and plasticity is well-motivated. Defining “skill neurons” through goal-oriented correlation is creative, interpretable, and biologically inspired.
- The method is conceptually simple yet effective, adding minimal complexity to standard SAC/PPO agents.
- Strong empirical evidence supports the claims. Meta-World, Atari, and DMC results consistently outperform baselines.
- Ablations and hyperparameter analyses (e.g., actor vs. critic masking) are thoughtfully designed.
- The paper is well-structured. Motivation, methodology, and evaluation flow logically, with clear notation and metric definitions (ASR, FM, FWT).
- Addresses a fundamental and persistent challenge in continual deep RL.
- NBSP’s interpretability and light implementation could make it broadly useful for future continual-learning research, including extensions to supervised or self-supervised setups.

**Weaknesses:**

- Experiments use only 3 seeds; more repetitions or confidence intervals would increase result reliability.
- The method’s dependence on the top-m% neuron threshold (m = 0.2), mask coefficient (α = 0.2), and averaging window size is not analyzed. A sensitivity study would strengthen robustness claims.
- The paper lacks variance or gradient-norm analyses explaining why masking high-score neurons stabilizes training.
- Atari and DMC results are summarized in tables but lack detailed learning curves or per-task analysis.
- The algorithmic description omits specifics such as how masks interact with target networks and the replay schedule.

**Questions:**

- How sensitive is NBSP to the hyperparameters?
- Have you measured gradient-norm variance or activation statistics to confirm reduced interference?
- Are both actor and critic masked at every step, or alternately? How does masking interact with target networks?
- What is the computational and memory overhead (FLOPs, wall-time, replay buffer size) compared to vanilla SAC?
- Have you examined the behavior of anti-correlated (“negative-score”) neurons?
- Would NBSP generalize to supervised continual learning or offline RL where reward signals differ?
- The paper excludes the final layer from neuron scoring. Why? Have you tested whether masking output neurons (for example, in the critic’s value head) hurts performance or stability?
-What failure modes did you observe, for example, situations where neuron correlation misidentifies unimportant units, or where masking accumulates and stalls learning?

**Details Of Ethics Concerns:**

There are no ethics concerns to report.

---

> ### Author Response · Authors · 2025-11-19
>
> We appreciate your valuable feedback on the experiment reliability, sensitivity study, learning curves, description, and generalization. And we carefully address your concerns in the following responses.
>
> Q1: seed
>
> A1: Thank you for your suggestion. To improve the reliability of our results, **we extend the number of random seeds from 3 to 5** for the methods(NBSP, vanilla SAC, and NE) on the task sequence (window-open  $\rightarrow$ window-close $\rightarrow$ drawer-open $\rightarrow$ drawer-close) with the highest performance variance in Meta-World. The updated results over five seeds are reported in the table below. While using more seeds improves statistical reliability, **the overall trends remain consistent and do not change our main conclusions.**
>
>   |             | ASR $\uparrow$ | FM $\downarrow$ | FWT $\uparrow$ |
>   | ----------- | -------------- | --------------- | -------------- |
>   | Vanilla SAC | 0.31 ± 0.08    | 0.81 ± 0.14     | 0.57 ± 0.27    |
>   | NE          | 0.55 ± 0.04    | 0.68 ± 0.08     | 0.89 ± 0.05    |
>   | NBSP        | 0.66 ± 0.10    | 0.49 ± 0.14     | 0.89 ± 0.08    |
>
>   Q2: sensitivity analysis
>
>   A2: Thanks for your suggestion.  We provide sensitivity analysis for the neuron proportion $m$ and the mask coefficient $\alpha$ in Appendix C.10.6, where NBSP demonstrates robustness across a broad range of values.
>
>   Regarding the averaging window size, **we additionally conduct experiments on the cycling task sequence (button-press-topdown → window-open), using multiple window sizes.**
>   As shown in the table below:
>
>   - When the window size is too small, the running averages of neuronal activations and performance become noisy, leading to inaccurate estimation of $\overline{a}$ and $\overline{q}$, which in turn hinders the correct identification of RL skill neurons.
>   - When the window size exceeds 50,000, NBSP becomes insensitive to the specific value, and performance remains consistently high.
>   - Since excessively large window sizes introduce unnecessary computational overhead, we choose 50,000 as a practical trade-off between estimation stability and efficiency.
>
>   | averaging window size | ASR $\uparrow$ | FM $\downarrow$ | FWT $\uparrow$ |
>   | --------------------- | -------------- | --------------- | -------------- |
>   | 5000                  | 0.81 ± 0.03    | 0.44 ± 0.08     | 0.98 ± 0.02    |
>   | 25000                 | 0.88 ± 0.03    | 0.3 ± 0.08      | 0.99 ± 0.01    |
>   | 50000                 | 0.95 ± 0.05    | 0.08 ± 0.12     | 0.98 ± 0.01    |
>   | 100000                | 0.94 ± 0.04    | 0.09 ± 0.11     | 0.98 ± 0.01    |
>   | 150000                | 0.95 ± 0.04    | 0.08 ± 0.11     | 0.98 ± 0.01    |
>
>
>
>   Q3: gradient-norm analyses
>
>   A3: Thank you for your suggestion. We would like to first clarify that, in our paper, **stability refers specifically to the agent’s ability to maintain performance on previously learned tasks**, which we evaluate using the FM metric. Therefore, gradient-norm statistics may not be suitable for evaluating this specific stability in our framework.
>
>   Nevertheless, we appreciate the reviewer’s point on the optimization stability during training and conducted an additional analysis. We measured gradient-norm statistics throughout training on the sequence (window-open → window-close → drawer-open → drawer-close). The results are reported in the table below, where values in parentheses denote task success rates, and the numbers before and after “±” denote the mean and variance of the gradient norms. We observe two phenomena:
>
>   1. **Lower gradient-norm variance under NBSP**:
>       NBSP consistently exhibits lower gradient-norm variance than vanilla SAC, indicating more stable optimization dynamics and reduced gradient interference during task transitions.
>   2. **Correlation between abnormal gradient norms and degraded task performance**:
>      Tasks with poor performance tend to coincide with anomalous gradient-norm values. This suggests a potential link between gradient-norm instability and failure to retain knowledge, which is an interesting future research direction beyond the scope of this work.
>
>   | Gradient-norm (success rate) | Window-open         | Window-close        | Drawer-open         | Drawer-close        |
>   | ---------------------------- | ------------------- | ------------------- | ------------------- | ------------------- |
>   | Vanilla SAC                  | 13.64 ± 4.10 (0.95) | 11.47 ± 7.45 (0.95) | inf (0.00)          | inf (0.35)          |
>   | NBSP                         | 9.21 ± 3.86 (0.95)  | 15.10 ± 5.62 (1.00) | 13.46 ± 4.71 (0.90) | 15.93 ± 5.87 (1.00) |

---

> ### Author Response · Authors · 2025-11-19
>
> Q4: Atari and DMC results
>
>   A4: Thanks for your question. We have added the experiment results of NBSP on Atari and DMC benchmarks in Appendix C.8 and C.9 in our revised submission.
>
>   Q5: description about specifics such as how masks interact with target networks and the replay schedule
>
>   A5: Thank you for pointing this out. NBSP consists of two key components: experience replay and gradient masking, and they operate independently.
>
>   1. Interaction with Replay Schedule
>      **The replay schedule determines whether sampled transitions belong to the current or previous tasks**. This affects only which data is fed into the update step.
>      In contrast, **gradient masking controls how the network parameters are updated, regardless of the data source.**
>      Therefore, the masking mechanism is orthogonal to the replay schedule. No matter which transitions are replayed, the corresponding parameter updates are always masked according to NBSP.
>   2. Interaction with Target Networks
>      Gradient masking is applied only to the online networks, where gradients are computed and parameters are updated.
>      The target networks are updated using EMA from the masked online networks.
>      Thus, the target networks automatically inherit the masked updates through the soft-update mechanism.
>      **No additional masking operation is required for target networks.**
>
>   3. Masking at Each Update Step
>      **During each gradient update, backward propagation computes the raw parameter update $\Delta W^{(l)}_{:,j}$.**
>      NBSP applies the mask to selectively block updates to RL Skill neurons of the actor and online critic network.
>
>   Q6: computational and memory overhead
>
>   A6: Thank you for the question. NBSP introduces minimal overhead beyond vanilla SAC.
>
>   1. Computation / Wall-time Overhead
>
>      - **Neuron identification is performed only once after each task is completed, not during training.**
>        Across the three example tasks (button-press-topdown, window-open, window-close), this process takes approximately 10 minutes on average.
>
>      - During training, the mask is applied only as an element-wise multiplication to the computed parameter updates. **The masking operation introduces only one additional floating-point multiplication per parameter.** Therefore, the extra computational overhead is exactly equal to the number of model parameters, i.e.,
>        $$
>        \text{FLOPs} = \sum_i |W_i|.
>        $$
>        This is negligible compared to the forward/backward passes of SAC ( <1\% of total training FLOPs). In our experiment, this operation increases the update time of the optimizer from 0.0006437 to 0.0006513 seconds.
>
>      - NBSP does not compute or maintain importance scores during training; therefore, FLOPs during rollout and learning are nearly identical to vanilla SAC.
>
>   2. Memory Overhead
>
>      - NBSP uses a replay buffer of size 1e5, the same scale commonly used in standard SAC.
>
>      - **The additional memory overhead corresponds only to storing the previous-task transitions $D_{\text{pre}}$**, which is bounded by:
>        $ |D_{\text{pre}}| \times  (2 \cdot state_{dim} + action_{dim} + 2)$
>        accounting for states, next states, actions, rewards, and done flags.
>
>      - No extra networks or parameter copies are maintained beyond SAC’s existing actor/critic and target networks.
>
>   Overall, **NBSP adds negligible runtime cost and only moderate buffer-related memory**, while keeping the training pipeline essentially identical to vanilla SAC.
>
> Q7: anti-correlated (“negative-score”) neurons
>
>   A7: Thank you for the insightful question. We define anti-correlated (or “negative-score”) neurons as those whose activations are below their average when the agent performs well, or above their average when the agent performs poorly. Conceptually, this is analogous to inhibitory units in biological neural systems, which can still play functional roles despite exhibiting activity patterns opposite to excitatory counterparts.
>
>   To better understand their behavior, we analyze the proportion of such neurons among all identified RL skill neurons. We found that anti-correlated neurons consistently account for roughly 10\% of the total RL Skill neuron set. **Although they represent only a small fraction, their presence is stable across tasks, suggesting that they may capture auxiliary or inhibitory aspects of task-specific representations**. A deeper investigation into the functional roles of these anti-correlated neurons is an interesting direction for future work.

---

> ### Author Response · Authors · 2025-11-19
>
> Q8: generalization to supervised continual learning or offline RL
>
>   A8: Thank you for the insightful suggestion. **Conceptually, NBSP can be extended to supervised continual learning and offline RL. However, the skill neuron identification module would need to be adapted**, as the current version relies on a Goal Metric (GM) defined in terms of success rate or average return in online RL.
>
>   - For supervised continual learning, a natural replacement for GM would be metrics derived from the supervised signal, such as task-specific loss or prediction accuracy. These metrics directly reflect the contribution of individual neurons to supervised task performance.
>   - For offline RL, where reward signals are fixed and no online interaction occurs, an offline-compatible measure such as the discounted return estimated from the dataset could serve as an alternative GM.
>
>   Exploring these adaptations is an interesting future direction. While the underlying NBSP framework is general, its effectiveness under these alternative paradigms would require further empirical validation.
>
>   Q9: final layer
>
>   A9: Thank you for your question. We exclude the final layer from neuron scoring for two main reasons:
>
>   1. The final layer contains very few neurons (e.g., a single output unit in the critic).
>       Masking such neurons would excessively constrain the network and directly damage its optimization capacity.
>   2. Final-layer activations correspond directly to network outputs.
>       These neurons inherently play a disproportionately large role in determining performance. As a result, in practice they are more likely to be identified as RL Skill neuron. Thus masking them is more likely to degrade the learning dynamics.
>
>   To validate this design choice, we conducted additional experiments where the final layer was included in the neuron identification and masking process. The results (shown in the table below) confirm our hypothesis. Overall performance deteriorates, primarily due to a substantial decrease in ASR,  and plasticity is harmed (lower FWT), as restricting output-layer neurons prevents the network from efficiently adapting to new tasks, and stability also worsens, because **once the final-layer neurons are masked, the network compensates by over-adjusting earlier RL skill neurons, inadvertently damaging previously acquired knowledge.**
>
>   |                        | ASR $\uparrow$ | FM $\downarrow$ | FWT $\uparrow$ |
>   | ---------------------- | -------------- | --------------- | -------------- |
>   | include the last layer | 0.76 ± 0.01    | 0.41 ± 0.02     | 0.87 ± 0.02    |
>   | exclude the last layer | 0.95 ± 0.05    | 0.08 ± 0.12     | 0.98 ± 0.01    |

---

> > ### Comment · Reviewer_QyWf · 2025-11-25
> >
> > Thanks for addressing my previous questions. There are some issues that should be fixed to improve clarity and presentation:
> >
> > * Line 59: The citation is incorrect; please revise it.
> > * Conclusion: Add a discussion of the paper’s limitations.
> > * Figure 8: The x-axis values appear wrong (e.g., row 1, column 3). Please correct them.
> > * Line 91: Please soften the claim: "We are the first to introduce the concept..."
> > * Line 469: "influenced by environment and context"; missing the article -> "influenced by the environment and context"
> > * Figure placement: Several figures are not located directly after their corresponding subsection headers. For instance, in Section C.8, Figures 13 and 14 should appear immediately after the subsection title. This improves the flow of the paper.
> > * Line 1382: Fix the formatting of “reasons:(1)”.
> > * Line 1501: Add a proper citation for PPO.
> > * Avoid using “And” repeatedly at the beginning of sentences (for example, in lines 1345–1347).
> > * As a suggestion, I would use a shorter version of the title, such as “NBSP: A Neuron-Level Framework for Balancing Stability and Plasticity in Deep RL.” It’s better to avoid split words in titles (e.g., Balanc-ing, Reinforce-ment Learning).
> >
> > If these changes are resolved, I will give this submission a 6, leaning toward acceptance.

---

> > > ### Author Response · Authors · 2025-11-26
> > >
> > > Thank you for your suggestions. We have carefully addressed the issues you raised and incorporated the corresponding revisions in our latest submission.

---

### Official Review · Reviewer_twf4 · 2025-11-01

**Soundness:** 4
**Presentation:** 3
**Contribution:** 4
**Rating:** 8
**Confidence:** 3

**Summary:**

The authors propose the NBSP framework to solve the stability-plasticity dilemma in DRL (Deep Reinforcement Learning). The method uses a goal-oriented strategy to identify "RL skill neurons" and combines adaptive gradient masking with experience replay to balance learning new skills while retaining old ones. Experiments on multiple benchmarks demonstrate that NBSP outperforms existing methods in multiple metrics such as ASR.

**Strengths:**

1. This paper introduces a neuron-level approach to address the stability-plasticity dilemma in DRL, proposing the concept of "RL skill neurons." While similar concepts have been discussed in other fields, their introduction to DRL is (to my knowledge) a novel contribution.
2. The experimental design is sound and thorough, featuring multiple benchmarks, diverse metrics, and a wide variety of baseline comparisons. Furthermore, the inclusion of extensive ablation studies to validate component effectiveness and explore the mechanisms of RL skill neurons significantly strengthens the paper's conclusions.
3. The paper is well-structured, clearly written, and utilizes well-designed figures and tables to effectively communicate the research outcomes.

**Weaknesses:**

1. The experimental setup is relatively simple:
    - Regarding the length of continual learning sequences, the paper tests on short task chains, focusing on lengths of 2 or 4 (at most), and lacks experiments on longer chains (e.g., over 10 Atari games).
    - Regarding task relatedness, the tasks tested are highly correlated, such as window-close/window-open in Meta-World and Cartpole-swingup/Cartpole-balance in DMC.
    - Finally, the tasks are relatively simple, raising questions about the method's generalizability to more complex environments.
2. The paper introduces a considerable number of hyperparameters:
    - Several hyperparameters are introduced for the RL skill neurons, and the two core techniques (gradient masking and experience replay) also bring their own.
    - **It is worth noting that the authors conducted sensitivity analyses, and the current results suggest the method is not overly hyperparameter-sensitive.**
    - However, it currently lacks a systematic selection criterion and relies on manual tuning, which raises doubts about its broader applicability to other tasks and algorithms.

**Questions:**

1. Could the authors provide more detailed implementation details for the baselines? For example, the original NE paper used the TD3 algorithm, but in this paper, it was implemented with SAC. Could the authors clarify how this algorithmic transition was handled to ensure a fair comparison?
2. While gradient masking is shown to significantly improve success rates, it resembles reducing the learning rate on certain neurons. Could the authors include experiments that reduce the learning rate across the entire network to demonstrate the necessity of selectively lowering it only on RL skill neurons?
3. The authors emphasize balancing stability and plasticity in DRL. However, both gradient masking and experience replay tend to enhance stability. Does this suggest that, for vanilla DRL algorithms, improving stability is more crucial for metrics like ASR? Alternatively, could NBSP be combined with other techniques that enhance plasticity, such as plasticity injection?

---

> ### Author Response · Authors · 2025-11-19
>
> We appreciate your valuable feedback on the experimental setup, hyper-parameter,  implementation details, and your acknowledgement of novelty, writing, and experimental design of our paper. We will address each of your comments and concerns in the following responses.
>
> Q1: the length of continual learning sequences
>
> A1: Thank you for your question. To further evaluate NBSP under longer task sequences, we additionally conduct experiments on a **10-task sequence** (button-press-topdown $\rightarrow$ window-close $\rightarrow$ door-open $\rightarrow$ drawer-close $\rightarrow$ drawer-open $\rightarrow$ door-close $\rightarrow$ button-press-topdown-wall $\rightarrow$ window-open $\rightarrow$ push $\rightarrow$ reach) in the Meta-World benchmark. As shown in the table below, NBSP continues to outperform vanilla SAC by a large margin, particularly in terms of stability, where the forgetting metric decreases from 0.79 to 0.32.
> We also compare NBSP with NE, which is the strongest baseline in our earlier Meta-World experiments. Although NE achieves a better stability–plasticity balance than vanilla SAC, it still lags behind NBSP on the longer 10-task sequence.
> These results demonstrate that **NBSP generalizes well beyond short task chains and is effective even in long-horizon task settings**.
>
> |             | ASR $\uparrow$ | FM $\downarrow$ | FWT $\uparrow$ |
> | ----------- | -------------- | --------------- | -------------- |
> | vanilla SAC | 0.27 ± 0.05    | 0.79 ± 0.07     | 0.52 ± 0.19    |
> | NE          | 0.58 ± 0.05    | 0.44 ± 0.04     | 0.64 ± 0.03    |
> | NBSP        | 0.66 ± 0.02    | 0.32 ± 0.06     | 0.74 ± 0.01    |
>
> Q2: task relatedness
>
> A2: Thank you for your question. Indeed, some tasks within the same sequence are correlated, such as (window-open, window-close) or (Cartpole Swingup, Cartpole Balance), which help demonstrate the effectiveness of NBSP in correlated task scenarios. Moreover, **we also conduct experiments on task sequences that contain uncorrelated tasks**, including (button-press-topdown, window-open), (button-press-topdown, window-close, door-open, drawer-close) in Meta-World (Table 1), (Pong, Bowling), (BankHeist, Alien) in Atari (Table 5), and (Walker Walk, Walker Stand) in DMC (Table 6). The results on these uncorrelated sequences further confirm the robustness and general effectiveness of NBSP across tasks with varying degrees of relatedness.
>
> Q3: task environment
>
> A3: Thank you for your insightful question. We would like to kindly argue that **our experiments cover three widely used benchmarks, including Meta-World, Atari, and DMC, which collectively span a broad spectrum of task complexity**.
>
> - Meta-World is one of the most common and challenging benchmarks for multi-task RL, featuring sparse rewards, diverse manipulation skills, and multi-task sequences used in prior works [1,2,3].
> - Atari, while simpler in visual dynamics, provides a discrete action space, allowing us to validate that NBSP generalizes beyond continuous-control settings.
> - DMC is built on the MuJoCo physics engine and introduces complex continuous control, rich dynamics, and perceptual disturbances. These characteristics make DMC a widely recognized and challenging benchmark in deep RL studies [1,4,5].
>
> Across all three benchmarks with varying difficulty levels, reward structures, and action spaces, NBSP consistently demonstrates strong performance, indicating its generalizability to diverse and complex environments.
>
> [1] Ahn H, Hyeon J, Oh Y, et al. Reset & distill: A recipe for overcoming negative transfer in continual reinforcement learning[J]. arXiv preprint arXiv:2403.05066, 2024.
>
> [2] He J, Li K, Zang Y, et al. Not all tasks are equally difficult: Multi-task deep reinforcement learning with dynamic depth routing[C]//Proceedings of the AAAI Conference on Artificial Intelligence. 2024, 38(11): 12376-12384.
>
> [3] He H, Bai C, Xu K, et al. Diffusion model is an effective planner and data synthesizer for multi-task reinforcement learning[J]. Advances in neural information processing systems, 2023, 36: 64896-64917.
>
> [4] Huang K, Shen L, Zhao C, et al. Solving continual offline reinforcement learning with decision transformer[J]. arXiv preprint arXiv:2401.08478, 2024.
>
> [5] Lewandowski A, Bortkiewicz M, Kumar S, et al. Learning Continually by Spectral Regularization[C]//The Thirteenth International Conference on Learning Representations. OpenReview. net, 2025.

---

> ### Author Response · Authors · 2025-11-19
>
> Q4: hyper-parameters
>
> A4: Thank you for raising this point. Our method involves only four hyper-parameters, and we provide a comprehensive sensitivity analysis in Appendix C.10.6. Based on these results, we summarize a practical and systematic tuning criterion as follows:
>
> 1. Stable performance within broad ranges.
>    Each hyper-parameter exhibits a wide “safe range” within which NBSP performs consistently well. Therefore, **tuning does not require fine-grained search**. The recommended ranges are provided in the table below.
>
>    - | Hyper-parameters | Range       |
>      | ---------------- | ----------- |
>      | m                | [0.15, 0.3] |
>      | $\alpha$         | [0.1, 0.3]  |
>      | $\|D_{pre}\|$    | [1e5, ∞)    |
>      | k                | [5, 13]     |
>
> 2. Key hyper-parameter for task variation.
>    Among the four hyper-parameters, **$m$ is the primary one that may vary across tasks**, as the number of RL skill neurons naturally depends on the skill complexity of each environment. In practice, tuning $m$ within [0.15, 0.3] is sufficient in our experiments.
>
> 3. Other hyper-parameters remain stable across benchmarks.
>    **The remaining three hyper-parameters require only minor adjustments within their recommended ranges**. In our experiments, they remain fixed across all benchmarks, and NBSP still achieves strong and stable performance, demonstrating their robustness and confirming that the method does not rely on brittle manual tuning.
>
> These findings show that NBSP is easy to tune, supported by clear sensitivity trends, and broadly applicable across diverse environments.
>
> Q5: implementation details for the baselines
>
> A5: Thank you for the question. To ensure a fair and controlled comparison, **all baselines were implemented on top of the same RL backbone (vanilla SAC)**, because different base algorithms (e.g., SAC vs. TD3) can yield significantly different performance on the same task. Using a unified backbone isolates the effect of each method and allows a direct comparison under identical training conditions.
>
> Below, we summarize how each baseline was adapted while strictly following its original design principles:
>
> - EWC
>   We incorporated the Fisher-weighted parameter change penalty into the SAC loss, following the original EWC formulation.
> - NPC
>   We applied neuron-wise consolidation using reduced learning rates based on the importance scores computed for SAC’s actor and critic networks.
> - ANCL
>   For each new task, we first train task-specific actor/critic networks using SAC, then distill them into the shared networks via the auxiliary loss defined in ANCL.
> - CoTASP
>   We use the official implementation for its base algorithm is SAC.
> - CReLU
>   We replaced the activation functions in SAC’s actor/critic networks with CReLU and adjusted hidden sizes according to the CReLU paper.
> - CBP
>   We substituted SAC’s optimizer with AdamGnT as proposed in CBP.
> - PI
>   For each new task, we freeze the previous critic and train a delta-critic (with the same architecture as SAC’s critic) following the original PI design.
> - NE
>   We retain the network expansion mechanism proposed in NE, dynamically growing the network from a small initial size to the full structure while training with SAC.
> - UPGD
>   We replaced SAC’s optimizer with the UPGD optimizer introduced in the original paper.
>
> For all baselines, we use the official GitHub implementations as references, and **we port their key components into a unified SAC framework**. This ensures that differences in performance are attributable to the baseline methods themselves, rather than discrepancies in backbone algorithms.

---

> ### Author Response · Authors · 2025-11-19
>
> Q6: learning rate
>
> A6:  Thank you for the insightful question. To verify that the effectiveness of adaptive gradient masking goes beyond simply reducing the learning rate, we additionally conduct experiments on the cyclic task sequence (button-press-topdown → window-open) comparing three settings:
>
> 1. Lowering the learning rate only on RL skill neurons
> 2. Lowering the learning rate on the entire network
> 3. Our proposed NBSP gradient masking
>
> The results are presented in the table below. We observe the following:
>
> - **NBSP achieves the best balance between stability and plasticity.**
>   NBSP adaptively applies different degrees of masking to RL skill neurons based on their score. This allows high-score neurons to be more strongly preserved while still enabling low-importance neurons to tune. A constant scaled learning rate cannot capture this differentiation and therefore underperforms NBSP.
> - **Lowering the learning rate on RL skill neurons only offers partial benefits but remains inferior.**
>   Treating all RL skill neurons uniformly ignores the heterogeneity in how different neurons contribute to task-specific behaviors. While this reduces interference to some extent, it still results in suboptimal stability–plasticity trade-offs.
> - **Lowering the learning rate across the entire network performs the worst.**
>   This merely slows down parameter updates without specifically protecting the neurons that encode task skills. As a result, RL skill neurons continue to be overwritten by new tasks, leading to high forgetting (FM) and degraded performance.
>
> Overall, these comparisons show that **the benefit of NBSP cannot be replicated by globally reducing the learning rate. The key advantage comes from adaptive and selective protection of RL skill neurons.**
>
> |                                    | ASR $\uparrow$ | FM $\downarrow$ | FWT $\uparrow$ |
> | ---------------------------------- | -------------- | --------------- | -------------- |
> | Lower the lr of RL Skill neuron    | 0.86 ± 0.07    | 0.27 ± 0.15     | 0.97 ± 0.02    |
> | Lower the lr of the entire network | 0.71 ± 0.12    | 0.68 ± 0.34     | 0.98 ± 0.01    |
> | NBSP                               | 0.95 ± 0.05    | 0.08 ± 0.12     | 0.98 ± 0.01    |
>
> Q7: stability and plasticity
>
> A7: Thank you for the question. In our experimental setting, we observe that vanilla SAC naturally exhibits strong plasticity but suffers from poor stability, leading to high forgetting across cyclic tasks. **NBSP is therefore designed primarily to improve stability. However, our adaptive gradient masking mechanism explicitly preserves a certain degree of tunability for RL skill neurons, preventing over-freezing and thus avoiding the plasticity collapse**. Therefore, our results do not imply that stability is inherently more important for ASR; rather, **they show that achieving a proper balance is crucial, and over-emphasizing stability alone can harm ASR by severely reducing plasticity, which could be observed in methods such as NPC**.
>
> To further examine the compatibility of NBSP with methods that explicitly enhance plasticity, we conducted an additional experiment on the task sequence (button-press-topdown → window-open) by integrating reset technique[6]. We test two variants: resetting targeting at the entire network, and resetting targeting at only the non–RL skill neurons. The results (see table below) show that combining reset with NBSP does not improve ASR. In fact, both variants degrade performance, primarily due to loss of previously acquired knowledge, which reduces stability. Moreover, resetting RL skill neurons (implicitly occurring in resetting targeting at the entire network) causes the largest stability drop, confirming that RL Skill neurons indeed encode essential skills for each task.
> **These results suggest that NBSP cannot simply be combined with techniques that enhance plasticity without considering their impact on stability. Designing a synergistic combination that preserves essential skill neurons while enabling additional plasticity is an interesting future direction.**
>
> |                                          | ASR $\uparrow$ | FM $\downarrow$ | FWT $\uparrow$ |
> | ---------------------------------------- | -------------- | --------------- | -------------- |
> | Reset targeting at the non-RL Skill neurons | 0.91 ± 0.05    | 0.14 ± 0.14     | 0.99 ± 0.00    |
> | Reset targeting at the entire network       | 0.75 ± 0.01    | 0.64 0.01       | 0.98 ± 0.01    |
> | NBSP                                     | 0.95 ± 0.05    | 0.08 ± 0.12     | 0.98 ± 0.01    |
>
>
> [6] Nikishin E, Schwarzer M, D’Oro P, et al. The primacy bias in deep reinforcement learning[C]//International conference on machine learning. PMLR, 2022: 16828-16847.

---

> > ### Comment · Reviewer_twf4 · 2025-11-23
> >
> > I thank the authors for their detailed responses and clarifications.
> >
> > **Regarding Q3**: I appreciate the authors' clarification regarding the selection of benchmarks; I agree that they are indeed standard and mainstream choices. However, even within these benchmarks, there exist significantly more complex tasks. For instance, tasks like `coffee-pull` and `push` in Meta-World present considerable challenges, as do harder Atari games like `Seaquest` and `Ms. Pacman`. Acknowledging that this work primarily builds upon the SAC algorithm, covering these highly complex tasks might fall outside the current scope. Nevertheless, I remain very interested in whether NBSP can be generalized to more complex algorithms and challenging tasks, which could serve as a valuable direction for future work.
> >
> > **Regarding Q6**: I appreciate the additional experimental results, which demonstrate that simply reducing the global learning rate is less effective than targeting only the RL skill neurons. I believe this further substantiates the validity of the NBSP method. However, could the authors please clarify the specific learning rate value used in this control experiment? Was it scaled by a factor of 0.2 relative to the original learning rate, or was another value chosen?

---

> > > ### Author Response · Authors · 2025-11-24
> > >
> > > Q3: generalization to more complex algorithms and challenging tasks
> > >
> > > A3: Thank you for the insightful suggestion. Indeed, vanilla SAC has limited performance on highly challenging tasks such as coffee-pull in Meta-World. Since our current work is built upon SAC, extending NBSP to these extremely difficult tasks would require integrating it with more advanced and expressive RL algorithms like TD3, HIRO.
> > >
> > > We agree that evaluating NBSP on stronger backbones and more complex tasks is an important and valuable direction. As part of our future work, we plan to apply NBSP to more powerful algorithms and conduct experiments on these harder environments to further validate its generalization ability and robustness.
> > >
> > > Q6:  learning rate value
> > >
> > > A6: Thank you for your question. In this control experiment, we scaled the original learning rate by a factor of 0.2 × 0.1, resulting in a value that matches the magnitude of the mask value based on the score of neurons. This ensures a fair comparison, as the learning rate adjustment operates at a similar scale to the neuron-level gradient masking applied in NBSP.

---

> > > > ### Comment · Reviewer_twf4 · 2025-11-26
> > > >
> > > > I thank the authors for their clarifications, my concerns have been addressed and I'll maintain my score of 8.

---

### Author Response · Authors · 2025-12-03

First of all, we sincerely thank all reviewers for their diligent reviews and constructive feedback. We are encouraged that all reviewers acknowledged the novelty, writing quality, and empirical validation of our work.

Following the reviewers' suggestions, we have primarily conducted the following additional experiments:

- As suggested by Reviewer twf4:
  - We compared the effect of lowering the learning rate with the gradient masking of RL skill neurons.
  - We examined the compatibility of NBSP with the reset technique, which explicitly enhances plasticity.
- As suggested by Reviewer QyWf:
  - We added a sensitivity analysis of the averaging window size.
  - We measured gradient-norm statistics throughout training.
  - We validated the design choice of excluding the final layer from neuron scoring.
- As suggested by Reviewer 4tP5:
  - We provided empirical evidence supporting our statements on gradient masking and experience replay.
  - We incorporated experience replay into several baseline methods to further substantiate NBSP’s advantage.
- As suggested by Reviewer t1NL:
  - We analyzed whether neurons that satisfy $(a > \bar{a}, q > \bar{q})$ also tend to satisfy $(a < \bar{a}, q < \bar{q})$ during the neuron-identification process.
- As suggested by Reviewer twf4, 4tP5:
  - We conducted additional experiments on a 10-task sequence to further evaluate NBSP’s generalization to longer task sequences.
- As suggested by Reviewer QyWf, t1NL:
  - We increased the number of random seeds from 3 to 5 for the Meta-World task sequence with the highest performance variance.

In addition to these supplementary experiments, we have also provided detailed explanations to address the reviewers' concerns:

- As suggested by Reviewer twf4:
  - We clarified experimental details, including task relatedness and task environments.
  - We summarized a practical and systematic tuning criterion.
  - We added implementation details for the baselines.
  - We clarified how NBSP balances stability and plasticity in DRL.
- As suggested by Reviewer QyWf:
  - We clarified how masks interact with target networks and the replay schedule.
  - We explained the computational and memory overhead.
  - We further analyzed anti-correlated (“negative-score”) neurons.
  - We clarified the generalization of NBSP to supervised continual learning and offline RL.
- As suggested by Reviewer 4tP5:
  - We explained the selection of benchmarks.
  - We clarified why experience replay and neuron masking are complementary.
- As suggested by Reviewer t1NL:
  - We clarified the neuron score function.
  - We explained the hyperparameter settings of other baselines.
  - We emphasized the distinction between CRL and DRL.
  - We introduced the generalization of NBSP to continuing settings.

We hope these additional experiments and clarifications could further strengthen our paper and satisfactorily address all raised concerns. All corresponding updates have been incorporated into the revised submission. We are also encouraged that our rebuttal has satisfactorily addressed many concerns of the reviewers: Reviewers QyWf and 4tP5 indicated their willingness to raise their scores from 4 to 6, Reviewer twf4 maintained a score of 8, and Reviewer t1NL is continuing further discussion. Thank you again for your valuable time and expertise.

Lastly, we would also like to thank all ACs and PCs for their significant efforts and contributions.

Best Regards,

Authors of Paper 15129

---

### Meta-Review · Area_Chair_MY1V · 2026-01-05

**Summary:**

I am recommending this paper to be rejected, despite two reviewers stating they would raise their scores after discussion. I am starting with my recommendation because it is maybe unexpected, and I do not want to write a paragraph that hints the paper will be accepted, only to then suddenly present a turn of events.

This paper proposes identifying neurons correlated with good performance on specific tasks and using gradient masking and replay to preserve them, as they appear critical for success across other tasks. Many concerns were raised by the reviewers, ranging from the length of tasks to the types of benchmarks used (e.g., MetaWorld, which is common in multi-task RL but not necessarily in the continual RL literature). The authors have addressed many of the concerns, and two reviewers have stated they would raise their scores from 4 to 6.

The main problem with the paper, however, is that most of its claims are backed by large tables of results, where the proposed methods are bolded to indicate they are superior to alternative baselines. However, these results are backed up by 5? (initially there were 3?) seeds, which is something quite hard to do. One could argue that the spread of results across many tasks is also indicative of something, but in this case, the paper should do something akin to a sign test to back up those claims. Instead, we have things such as the last row of Table 2, where the interval in the third column likely overlaps with the fourth and fifth, and that is not addressed. Maybe even more importantly, the baseline methods did not have their hyperparameters tuned; the default ones were used. This is ok when one is evaluating their method in the setting in which those baselines were introduced, since it is the authors' responsibility to put forward the best performance. However, when one is transposing a method to a different benchmark, it falls on their shoulders the responsibility to do justice to those baselines. It doesn't seem to have been the case here.

I also want to point out that the idea of having individual neurons tied to individual tasks might face a scalability issue if one is to think of an agent with a very long lifespan. This is maybe what was being hinted at by the reviewers when complaining about the short length of tasks (and even 10 wouldn't necessarily address such a concern, given the mechanics of the proposed method).

I want to conclude by saying I am sorry for the recommendation I feel forced to make. I acknowledge this paper would likely be accepted under a different AC, but the paper swaps and the need to look more closely at each submission have led to this outcome.

**Reviewer Concerns:**

- _The experimental setup has a few tasks, and the tasks are too similar to each other._

 	The authors argue that there are tasks that are different, and that they have used standard benchmarks in the field.

 - _The proposed approach adds many new hyperparameters to the algorithm._

 	Authors argue that their method is robust to such choices and that they mostly don't matter.

 - _Results were provided for only three seeds._

 	As expected (and usual), the authors raised the number of seeds to 5.

 - _Method is evaluated only in RL tasks, not supervised learning ones._

 	I personally think this is totally fine, as this is what the paper set itself to do.

 - _Hyperparameters of the baselines were not tuned._

 	This is a very important criticism. The authors state that they used the default values introduced by the baselines, but those values were introduced in a different context. It is wrong to make major claims about one method outperforming the other, given that the baselines were not properly tuned.

 - _Details about the analysis and the method are missing._

**Reviewer Scores:**

- Reviewer twf4: Explicitly stated they would maintain their original score of 8.
 - Reviewer QyWf: Initially gave the paper a 4, but explicitly stated that they would give the paper a 6 if the issues they raised were addressed.
 - Reviewer 4tP5: Initially gave the paper a 4 but raised their score to 6 after extensive discussion with the reviewers.
 - Reviewer t1NL: Initially gave the paper a 4 and would likely have kept the 4.

---

### Decision · Program_Chairs · 2026-01-26

Reject